# PET-measured human dopamine synthesis capacity and receptor availability predict trading rewards and time-costs during foraging

Angela M. Ianni [1,2,4] ✉, Daniel P. Eisenberg [1], Erie D. Boorman[2], Sara M. Constantino [3,5,6,7], Catherine E. Hegarty[1], Michael D. Gregory [1], Joseph C. Masdeu [1,8,9], Philip D. Kohn[1], Timothy E. Behrens [2,10] & Karen F. Berman[1,10]

Foraging behavior requires weighing costs of time to decide when to leave one reward patch to search for another. Computational and animal studies suggest that striatal dopamine is key to this process; however, the specific role of dopamine in foraging behavior in humans is not well characterized. We use positron emission tomography (PET) imaging to directly measure dopamine synthesis capacity and $D_1$ and $D_{2/3}$ receptor availability in 57 healthy adults who complete a computerized foraging task. Using voxelwise data and principal component analysis to identify patterns of variation across PET measures, we show that striatal $D_1$ and $D_{2/3}$ receptor availability and a pattern of mesolimbic and anterior cingulate cortex dopamine function are important for adjusting the threshold for leaving a patch to explore, with specific sensitivity to changes in travel time. These findings suggest a key role for dopamine in trading reward benefits against temporal costs to modulate behavioral adaptions to changes in the reward environment critical for foraging.

Foraging is a type of reward-guided behavior that is essential for survival and conserved across species. In contrast to explore-exploit decision-making paradigms where the decision is to repeat (exploit) a familiar action or explore a new one, the important choice when foraging is whether to engage with the current environment or leave and search elsewhere. In foraging behavior, rewards tend to occur in clusters (patches) that deplete over time as the reward is consumed. In a canonical example of an animal searching for food, the reward patch is a tree filled with berries and the foraging decision involves when to leave one tree to search for another[1–3]. In modern humans, foraging decisions typically involve how to spend time and when to switch from one rewarding activity to another, ranging from deciding when to leave one sight to visit another when exploring a new city on vacation to browsing social media[4].

Foraging behavior can be quantified using the marginal value theorem (MVT)[5], and experimental evidence supports the notion that

[1]Clinical & Translational Neuroscience Branch, National Institutes of Mental Health, Intramural Research Program, National Institutes of Health, Bethesda, MD, USA. [2]Wellcome Centre for Integrative Neuroimaging, University of Oxford, Oxford, United Kingdom. [3]Department of Psychology, New York University, New York, NY, USA. [4]Present address: Department of Psychiatry, University of Pittsburgh, Pittsburgh, PA, USA. [5]Present address: School of Public Policy and Urban Affairs, Northeastern University, Boston, MA, USA. [6]Present address: Department of Psychology, Northeastern University, Boston, MA, USA. [7]Present address: School of Public and International Affairs, Princeton University, Princeton, NJ, USA. [8]Present address: Houston Methodist Institute for Academic Medicine, Houston, TX, USA. [9]Present address: Weill Cornell Medicine, New York, NY, USA. [10]These authors contributed equally: Timothy E. Behrens, Karen F. Berman. ✉e-mail: ianniam@upmc.edu

this theorem closely describes the behavior of wild animals[1,6] and humans performing a computer-based foraging task[2]. According to the MVT, animals should leave a patch of rewards in search of a new one when the reward rate in the current patch falls below the average reward rate for the environment[5]. In other words, when the current setting is less rewarding than the average experienced by the animal, it is optimal to leave and search for something that is likely to be better. In many ecological settings and in our experiment, the average reward rate for the environment depends on two factors: (1) the travel time between patches of rewards and (2) the reward depletion rate within each patch of rewards. For instance, a tourist who is sightseeing in a new city might remain longer at any given attraction if the next location is further away or if the thrill of experiencing the current attraction dissipates more slowly. Therefore, both travel time and reward depletion rate should be accounted for when deciding when to leave a reward patch. Of note, experimental data from humans, nonhuman primates, and other animals has shown a deviation from the MVT such that there is a consistent bias to stay in reward patches longer than optimal[1,2,7–9]. This could reflect factors not accounted for in the MVT such as preference for immediate over delayed rewards, risk of predation during travel between reward patches, activities that occur simultaneously during foraging (e.g. parental care, searching for mates), and varied nutritional states (e.g. hungry vs. satiated)[8,9]. However, past studies have shown that measuring the relative change in patch-leaving threshold between reward environments controls for individuals' bias to stay and more closely reflects optimal behavior modeled with the MVT[3,7].

Prior work has described optimal foraging behavior and the involved neural circuitry. A seminal study in macaques showed that neural activity in the anterior cingulate cortex (ACC) plays a key role in adjusting the patch-leaving threshold[1]. Specifically, the firing rate of neurons in the ACC increases the longer an animal stays in a reward patch until a specific threshold is reached. Longer travel times increased the threshold required for the patch-leaving decision, however, the underlying mechanism for how the threshold is set is not accounted for by ACC neural activity alone. A subsequent fMRI study in humans found that foraging behavior depends on the function of a widespread circuit in the brain including the midbrain, striatum, medial prefrontal cortex, and ACC, with the ACC encoding the average value of the environment and cost of foraging[7]. While this human study revealed commonalities in the neural basis of foraging behavior across species, it did not use a patch-foraging paradigm and therefore was unable to provide insight into how the patch-leaving threshold is set in human foragers, and was agnostic to putative neurochemical mechanisms proposed in formal computational theory[10,11]. One possible mechanism for encoding information about the environment that subsequently modulates the threshold for leaving is through neuromodulation. Dopamine is a neuromodulator that is strikingly conserved across species and known to play a key role in both learning and modulation of circuits in the prefrontal cortex[12,13].

There is limited empirical evidence on the role of dopamine in foraging behavior in humans. Computational models predict that striatal tonic dopamine encodes the average reward rate of the environment[10] and, along with dopamine receptor activation, plays a role in weighing costs and benefits in the decision to exploit known reward sources or explore for new ones[11,14,15]. Tonic dopamine levels are predicted to vary as a function of the average reward rate of the current reward environment, increasing in rich environments when reward patches are abundant and located in proximity compared to poor environments when rewards are sparse and dispersed. Increased tonic dopamine is also thought to drive increased rate and vigor of response seen in animal studies[10], as well as modulate how benefits and costs of actions are represented at the time of choice[11,16,17]. Three studies in humans have supported the role of dopamine in foraging

behavior. In two studies of Parkinson's disease, patients off dopaminergic medications tended to wait longer than controls before leaving a reward patch, and dopaminergic medication partially ameliorated this impairment[18,19]. A pharmacological study in healthy controls found that administration of a $D_2$ agonist modulated foraging decisions in poor environments only[3]. However, we lack understanding of the role of $D_1$ receptors, and critically, the spatial localization of dopamine effects on foraging behavior.

While there have not been any human studies investigating the role of dopamine $D_1$ receptors in foraging behavior, computational models and work in animals and human genetics suggest that both $D_1$ and $D_2$ receptors are important for decisions that involve weighing costs and benefits and adjusting responses to maximize rewards. Specifically, there is a body of evidence supporting opposing learning effects mediated by $D_1$ and $D_2$ receptors facilitating approach and avoidance learning, respectively[16,17]. Tonic dopamine at the time of choice is thought to modulate the $D_1$ and $D_2$-mediated action values to differentially affect representations of benefits and costs. A combined pharmacological and PET study in monkeys revealed that blockade of either $D_1$ or $D_2$ receptors reduced the impact of reward and increased delay discounting through a synergistic effect[20]. Furthermore, rat and human genetic studies have supported differential roles of $D_1$ and $D_2$ receptor function in learning from positive and negative outcomes in comparative decision making tasks[21,22]. In a continuous-space explore-exploit task, genetic variation in $D_1$ and $D_2$ receptor expression was associated with complimentary roles in adjusting response times to maximize rewards, with $D_1$ receptors implicated in speeding up and $D_2$ receptors associated with slowing down responses[23]. However, a more classic explore-exploit multi-arm bandit task did not find any changes in exploration or exploitation behavior with exogenous administration of a $D_2$ receptor antagonist[24]. While there have been no prior studies comparing both $D_1$ and $D_2$ receptor function during foraging, there are similarities in the decision-making process that suggest both receptor types may be implicated in foraging behavior as well. For example, both foraging and delay discounting decisions require weighing immediate rewards against the cost of lost time. In addition, like comparative decision making, foraging involves integrating both positive and negative feedback in the form of evaluating whether a reward was received or not at every unit of time. Finally, both explore-exploit and foraging decision-making involve balancing the benefits of exploiting or sticking with a familiar option (e.g. staying in the current reward patch) with exploring alternative options (e.g. leaving for a new reward patch).

To gain a comprehensive understanding of the role of dopamine in foraging behavior in humans, we used positron emission tomography (PET) imaging to directly measure dopamine presynaptic synthesis capacity and dopamine $D_1$ and $D_{2/3}$ receptor availability separately in healthy adults combined with a well-validated[2,19], computer-based task designed to measure foraging decision-making. Based on the non-human primate study showing that leave decisions occurred when ACC neural firing rate reached a set threshold that was modulated by travel time[1] as well human studies implicating both exogenous dopamine administration and $D_2$ receptor agonism in patch leaving threshold changes[3,18,19], we predicted that dopamine function plays a key role in adjusting foraging behavior in response to environmental changes. Specifically, we hypothesized that individuals with higher dopamine synthesis capacity and receptor availability would enact greater adjustments in the patch leaving threshold as the reward environment changed. Preclinical work in rodent models has suggested that presynaptic synthesis capacity may be related to constitutive dopamine neuron population activity (or average number of spontaneously active dopamine neurons), which is thought to influence tonic dopamine efflux[25–27]. Furthermore, based on studies of related reward-guided decision-making tasks[20–22], we hypothesized that both $D_1$ and $D_{2/3}$ receptors would be

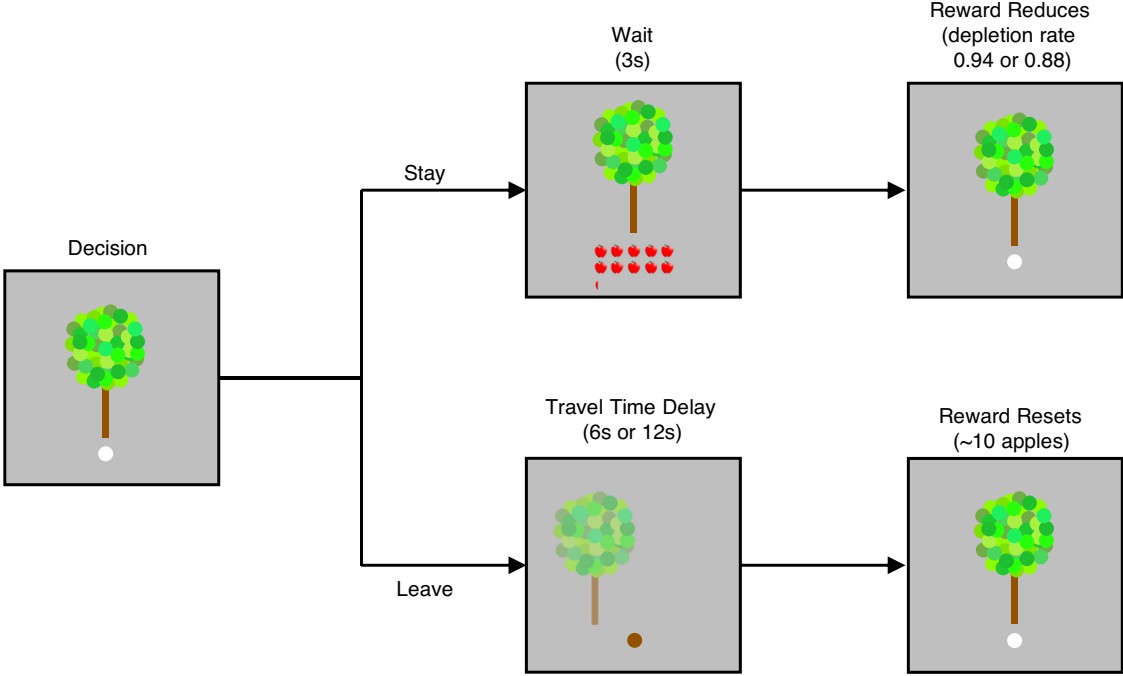

**Fig. 1 | Patch foraging task schematic.** Participants decided whether to stay at the current tree and harvest it for apples or leave and search for a new tree. If they decided to stay, they would receive a certain number of apples, shown below the tree, which was later translated to monetary reward added to their compensation. The number of apples remaining in the tree would then decrease according to a set depletion rate. Subjects would then make the stay or leave decision again. There were infinite new trees available. If participants decided to leave, they had to endure a travel time delay until they reached a new tree. This task was completed in four different reward environments, varying in travel time, which was either long (12 s) or short (6 s), and reward depletion rate, which was either steep (0.88 times previous reward) or shallow (0.94 times previous reward). Each block lasted 6.5 min and travel time and depletion rate remained constant throughout the block. Blocks were presented in random order across participants. Task adapted for current experiment in collaboration with Sara Constantino.

important for patch-leaving threshold adjustments through synergistic or complimentary effects. Lastly, while there have been no prior studies spatially localizing dopamine effects in foraging humans, we predicted that dopamine function in the ventral striatum would be particularly important for foraging decisions based on computational and animal studies highlighting its role in encoding the average reward rate of the environment[10,28] and evaluating the opportunity cost of time[29].

## Results

### Participants

Fifty-seven healthy adults (29 females, aged 21.4–57.6 years, mean age 35.4, standard deviation 9.9) were recruited from the local community. All studies were completed at the National Institutes of Health Clinical Center and were approved by the Combined Neuroscience Institutional Review Board and the National Institutes of Health Radiation Safety Committee. Participants completed written informed consent and were screened by a clinician-administered history and physical exam, routine laboratory testing, and structural MRI read by a neuroradiologist to rule out confounding medical and psychiatric disorders.

### Dopamine PET neuroimaging measures

These volunteers completed a multi-tracer PET study of dopamine synthesis and receptor availability at a separate time from completing a computer-based patch foraging task. The PET study consisted of three scans, which a subset of the participants completed, including [18F]-FDOPA (51 subjects) to assess dopamine presynaptic synthesis capacity, [11C]-NNC112 (45 subjects) to assess dopamine $D_1$ receptor binding potential, and [18F]-Fallypride (42 subjects) to assess dopamine $D_{2/3}$ receptor binding potential. Dopamine synthesis capacity was measured over a 90-min period of rest and therefore presumably

reflects a basal tonic dopamine synthesis rate[27]. Thirty-seven individuals completed all three PET scans.

### Patch-foraging task

To quantify foraging behavior, we used a well-validated computer-based patch-foraging task[2,19]. The foraging task was completed during a behavioral testing session that included one other probabilistic decision-making task, either before or after an MRI scan for a different decision-making task. The order of the behavioral tasks was randomly counterbalanced across participants and the order of behavioral testing session and MRI scan was determined by logistical constraints. The behavioral task was collected on a separate day from the PET scans with a median of 9.0 months between the behavioral task and the [18F]-FDOPA and [18F]-Fallypride scans and 5.5 months between the behavioral task and the [11C]-NNC112 scan. Sensitivity analyses controlling for time between behavioral task and PET scan are included in the Supplementary Materials as well as evidence of test-retest reliability for the behavioral measures from an independent sample (see Supplementary Fig. 8). The results from these analyses were unchanged from the original results.

Participants aimed to collect as many apples as possible from apple trees. To achieve this goal, they decided whether to stay at a given tree and harvest it for apples or leave and search for a new tree (see Fig. 1). If participants stayed at a tree, they received a number of apples, later converted to monetary reward. The apples within a tree depleted over time according to a set depletion rate. If participants decided to leave, they had to endure a travel time delay until they reached a new tree. We measured the threshold at which individuals decided to leave a depleting apple tree across four different experimental reward environments that varied in their average reward rates, here controlled by travel time between trees (long or short), and depletion rate of apples within a tree (steep or shallow). Recall that the MVT predicts that participants will leave a patch when reward depletes below the average

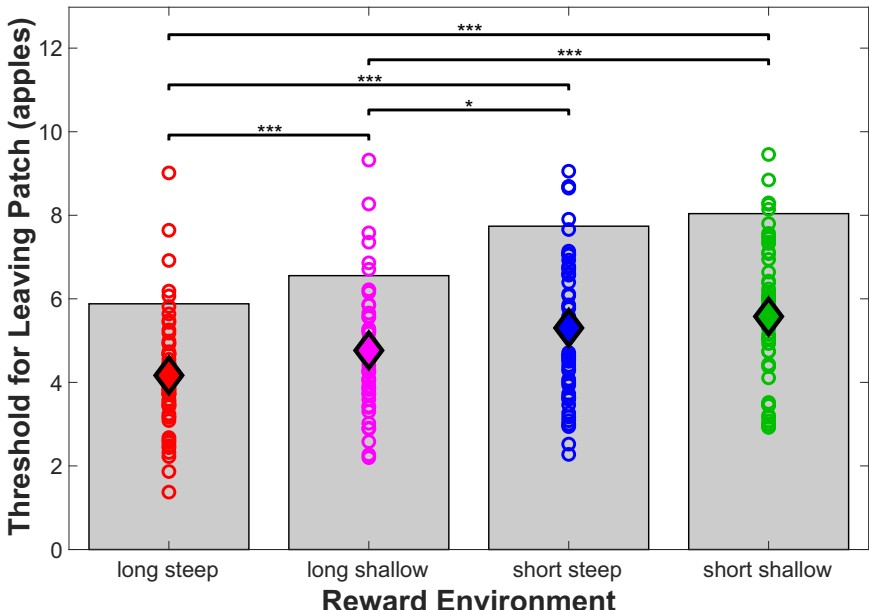

**Fig. 2 | Reward patch leaving thresholds across reward environments.** The average threshold for leaving a reward patch is shown for the group (filled diamonds) and for each participant (open circle, $n = 56$) for each reward environment. The optimal thresholds as calculated from the marginal value theorem are indicated with the gray bars (5.88, 6.56, 7.74, 8.04 for long steep, long shallow, short steep, and short shallow reward environments, respectively). The group average thresholds are denoted with the colored diamonds. Two factor repeated measures ANOVA revealed independent effects of both decay rate and travel time ($p < 0.0373$ for decay rate and $p < 5.24e{-}6$ for travel time; $n = 56$ individual participants) but not the interaction ($p = 0.438$). Post-hoc paired t-tests revealed that the threshold for leaving a patch is significantly lower in the long-steep reward environment compared to the long-shallow ($p = 2.23e{-}4$), short-steep ($p = 9.16e{-}7$), and short-shallow ($p = 3.72e{-}7$) environments. The threshold for leaving a patch was lower in the long-shallow reward environment compared to the short-steep ($p = 0.0147$) and short-shallow ($p = 2.76e{-}4$) environments. Results from post-hoc paired t-tests are indicated on the figure as follows: $*p < 0.05$, $***p < 0.001$. Source data are provided as a Source Data file.

reward rate in the environment. When the MVT is applied to the foraging paradigm used in this experiment[2,19], the optimal leaving rule is to search for a new tree when the expected reward from the next harvest, $E_{s_{i+1}}$, is less than the time spent harvesting, $h$, times the average reward rate, $\rho$. The state of the tree ($s_i$) is equal to the received reward and rewards deplete according to a depletion rate, $\kappa$; therefore, the expected reward from the next harvest is equal to the depletion rate, $\kappa$, times the received reward, $s_i$. Taken together, the MVT predicts that the optimal leave decision occurs when $\kappa s_i < \rho h$. Hence, the reward rates at the time of leaving (leaving threshold) should differ between environments with different average rates of reward and should be affected by depletion rate and time costs.

To assess individual differences in sensitivity to the foraging reward environment, we calculated the difference in leaving thresholds between the reward environments with the highest (short travel time and shallow depletion rate) and lowest (long travel time and steep depletion rate) average reward rates. This was chosen as our primary behavioral measure of interest based on past studies showing that measuring the relative change in patch leaving threshold between reward environments controls for individuals' bias to stay and more closely reflects optimal behavior modeled with the MVT[1,3]. We tested various dynamics of the patch leaving threshold, which are included in the Supplementary Materials (Supplementary Figs. 1–7) and did not affect our conclusions. We also measured three secondary behavioral parameters of interest. To investigate individual differences in sensitivity to changes in travel time, we calculated the difference between the average leaving threshold for the long and short reward environments. Likewise, we calculated sensitivity to changes in depletion rate by taking the difference between the average leaving threshold for the steep and shallow environments. Due to the factorial nature of the design, these two factors are unconfounded. Lastly, to assess for relative changes in response invigoration, we measured the change in reaction time between the most and least rewarding environments.

## Participants follow optimal foraging pattern

Participant behavior generally followed the same pattern as predicated by the MVT wherein subjects made foraging decisions on the basis of the average environment value (see Fig. 2). Optimal leaving thresholds calculated from the average of 100,000 simulations of the MVT were 5.88, 6.56, 7.74, 8.04 for the long steep, long shallow, short steep, and short shallow reward environments, respectively (shown as gray bars in Fig. 2). A two-way ANOVA with factors travel time and depletion rate revealed main effects of both environmental factors on leaving threshold ($p < 0.0373$ for decay rate and $p < 5.24e{-}6$ for travel time) but not the interaction ($p = 0.438$). Thresholds for leaving a reward patch reflected both travel time ($t = 5.614$, $p = 6.704e{-}7$) and depletion rate ($t = 2.914$, $p = 0.0052$). Post-hoc t-tests revealed significant differences for all pairs of reward environments except the short steep and short shallow environments (greatest difference found between short shallow and long steep reward environment leaving thresholds: $t = 5.774$, $p = 3.720e{-}7$; see Fig. 2). However, participants tended to stay in all patches for longer than optimal ($t = 10.386$, $p = 0.0019$), consistent with multiple prior studies[2,3,19,30] and likely reflecting factors not included in the MVT and our analyses such as risk of predation when traveling or asymmetrical learning rates[30]. Notably, behavioral sensitivity to travel time and decay rate parameters were uncorrelated ($r = 0.137$, $p = 0.314$). Some participants were more affected by travel time, others by reward depletion.

## Ventral striatal $D_1$ and widespread striatal $D_{2/3}$ receptor availability are associated with changes in patch leaving threshold

To test our hypothesis that striatal $D_1$ and $D_{2/3}$ dopamine receptor availability modulates the threshold at which humans leave patches to explore, we first ran a voxelwise GLM for each PET modality including our primary behavioral measure of interest, change in patch-leaving threshold, with age and sex included as covariates of no interest. MNI-space voxelwise data were restricted to the dopamine-rich basal

**Table 1 | Voxelwise results for GLM regression with change in patch leaving threshold**

| Dopamine measure | MNI coordinates (mm) | Peak voxel t-stat | Region |
|---|---|---|---|
| $D_1$ receptor availability | (−25.5, 7.5, −10.5) | 3.27 | left ventral putamen |
|  | (22.5, 19.5, −9) | 3.08 | right ventral putamen |
| $D_{2/3}$ receptor availability | (−28.5, 6, −6) | 3.34 | left putamen |
|  | (24, 0, 0) | 3.09 | right putamen |
|  | (12, 18, −6) | 3.18 | right ventral caudate nucleus |
|  | (−15, 9, 9) | 2.99 | left dorsal caudate nucleus |
|  | (12, 3, 15) | 3.48 | right dorsal caudate nucleus |

Peak voxel stats are shown for all clusters meeting significance threshold of TFCE FWE corrected for multiple comparisons at a threshold of $p < 0.05$, two-sided, small volume corrected within the basal ganglia.

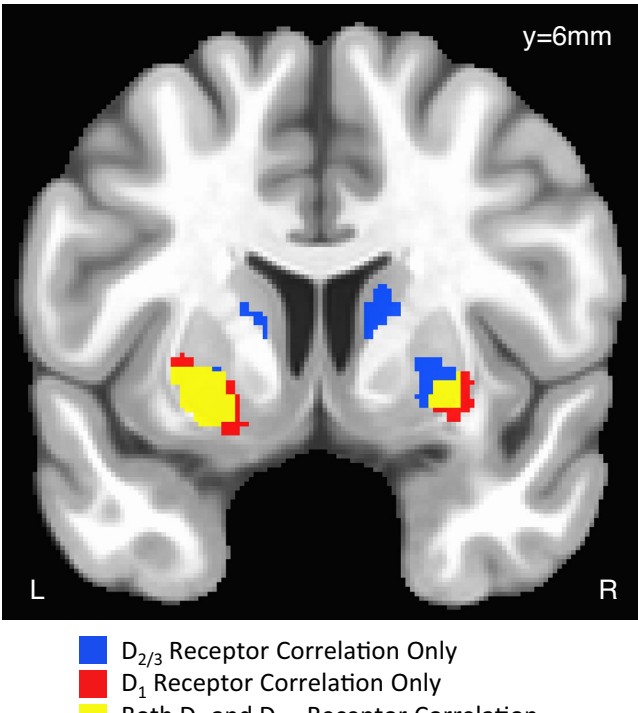

■ $D_{2/3}$ Receptor Correlation Only
■ $D_1$ Receptor Correlation Only
■ Both $D_1$ and $D_{2/3}$ Receptor Correlation

**Fig. 3 | Voxelwise results for linear regression of $D_1$ and $D_{2/3}$ dopamine receptor availability and total change in exit threshold.** Voxels that remain significant at a threshold of TFCE FWE corrected for multiple comparisons at a threshold of $p < 0.05$, two-sided, small volume corrected within the basal ganglia. Significant voxels for regression between total change in exit threshold and $D_1$-receptor availability ($n = 45$ individual participants) are colored red, $D_{2/3}$-receptor availability ($n = 40$ individual participants) are colored blue, and overlapping voxels significant for both $D_1$- and $D_{2/3}$-receptor availability are colored yellow.

ganglia. We found that the change in exit threshold was positively correlated with dopamine $D_1$ receptor availability in the bilateral ventral putamen and with dopamine $D_{2/3}$ receptor availability in the bilateral putamen, right ventral caudate nucleus, and bilateral dorsal caudate nucleus ($p_{FWE} < 0.05$; see Table 1 and Fig. 3). There were no significant correlations between the total change in patch-leaving threshold and presynaptic dopamine synthesis capacity at our threshold of $p_{FWE} < 0.05$.

**PCA analysis combining PET measures reveals four patterns of dopamine synthesis and receptor variability**
Next, we sought to use region of interest (ROI) data to replicate the voxelwise results and further investigate how the different contributors to exit threshold relate to patterns of dopamine

function. However, given that the three PET measures are not independent (e.g. ROI values between regions within each tracer are strongly correlated and dopamine receptors can be down- or up-regulated based on tonic dopamine levels[26]), constructing a linear regression that included all PET ROI measures would result in lost information. Principal component analysis (PCA) is ideally suited to capitalize on the unique multimodal design of this study and allow us to identify distinct patterns of variability across the three dopamine PET tracers. We first extracted data from five predetermined ROIs including the ACC, midbrain, ventral striatum, dorsal caudate nucleus, and dorsal putamen (see Fig. 4a). The midbrain and striatal regions were chosen because of their known strong dopaminergic innervation. Midbrain data was excluded for the $D_1$ receptor tracer ($[^{11}C]$-NNC112) due to low signal reliability in that region, evidenced by poor compartment model fit. The ACC was included as an additional ROI because of prior animal and human fMRI studies showing that neural activity in this region is key for foraging decision-making[1,7]. In addition, animal studies have shown that the supragenual ACC is the cortical region with the highest density of dopamine innervation[31] and dopamine projections to the ACC have been implicated in effort-based decision making[32,33].

Prior to running the PCA, we regressed out the effects of age and sex for all PET ROI measures and then normalized the residual values for each PET tracer ROI across subjects. We then compiled a matrix including the normalized PET ROI data from each region for all three tracers for the 37 subjects who completed all three PET scans. PCA on this matrix generated four patterns of consistent dopamine variation across subjects accounting for 74% of the total variance (see Fig. 4 for component weightings and associated eigenvalues). The first component yielded a pattern of high dopamine $D_1$ and $D_{2/3}$ receptor availability throughout the striatum. The second component reflects a pattern of high dopamine presynaptic synthesis capacity throughout the basal ganglia along with high striatal $D_{2/3}$ receptor availability in all regions except the midbrain. The third component contains a pattern of high $D_1$ receptor availability and dopamine presynaptic synthesis capacity in all regions coupled with low $D_{2/3}$ receptor availability. Finally, component 4 is mostly driven by high mesolimbic (ventral striatum and midbrain) dopamine presynaptic synthesis capacity and $D_1$ and $D_{2/3}$ receptor availability along with high presynaptic synthesis capacity and low $D_{2/3}$ receptor availability in the ACC. We assessed reliability of the PCA solution using an independent sample of 26 individuals and found that components 1 and 2 were stable across samples (component 1 between sample $r = 0.702$, $p = 0.005$; component 2 between sample $r = −0.559$, $p = 0.038$) whereas components 3 and 4 were not as robustly stable (component 3 between sample $r = 0.169$, $p = 0.563$; component 4 between sample $r = 0.067$, $p = 0.820$; see Supplementary Materials for additional details and results from bootstrap analysis).

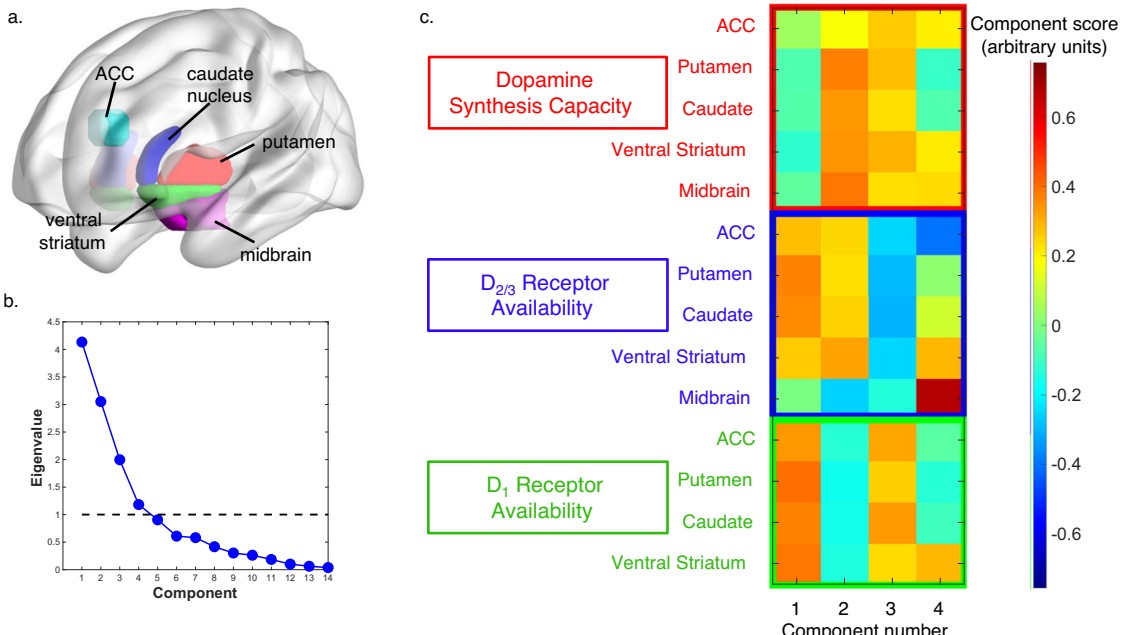

**Fig. 4 | Regions of interest (ROIs) and Dopamine PET principal component analysis (PCA) Results. a** ROIs for one example participant including the dorsal putamen (red), dorsal caudate nucleus (blue), ventral striatum (green) and dopaminergic midbrain (magenta). Anterior cingulate cortex (ACC) ROI (cyan) created as a 5 mm sphere around the MNI coordinates (−4, 32, 20) from Kolling et al. foraging search value peak voxel [7]. **b** Screen plot of eigenvalues for all components identified with dopamine PET PCA analysis. The cut-off line of one is shown as a dotted black line and the four components with eigenvalues greater than one were used for regression analyses with foraging behavioral parameters. **c** Component scores (arbitrary units) from the four PCA components with eigenvalues greater than 1. [18F]-FDOPA ROI values are shown in red (Dopamine Synthesis Capacity), [18F]-Fallypride values are shown in blue (D$_{2/3}$ Receptor Availability), and [11C]-NNC112 values are shown in green (D$_1$ Receptor Availability). Source data are provided as a Source Data file.

## ROI and PCA analyses replicate voxelwise results and reveals a pattern of mesolimbic and ACC dopamine function associated with changes in patch-leaving threshold

A linear regression with our primary behavioral measure of interest, total change in leaving threshold, as the dependent variable and the four dopamine PET PCA component scores as the independent variables, revealed two patterns of dopamine variability that correlated with change in patch-leaving threshold (regression model $F = 3.27$, $p = 0.0234$; see Fig. 5). First, we replicated the voxelwise results, showing that greater adjustment in the patch-leaving threshold was associated with widespread dopamine D$_1$ and D$_{2/3}$ receptor availability throughout the striatum (component 1; $t = 2.691$, $p = 0.0114$). Second, we identified a more localized pattern of dopamine effects on threshold changes encompassing high mesolimbic PET values in all three dopamine tracers as well as high presynaptic synthesis capacity and low D$_{2/3}$ receptor availability in the ACC (component 4; $t = 2.341$, $p = 0.0259$).

To aid in interpretation of the PCA analyses, we ran partial correlation analyses for each individual ROI, controlling for age and sex. We found that the ROI results were consistent with the voxelwise analyses. Specifically, total change in patch-leaving threshold was positively associated with D$_1$-receptor availability in the ventral striatum ($r = 0.378$, $p = 0.0123$) and there were positive trends in several other regions including D$_1$ receptor availability in the putamen and ACC, D$_{2/3}$ receptor availability in the ventral striatum, caudate nucleus, and putamen, and presynaptic synthesis capacity in the ACC (all $p < 0.1$; see Supplementary Materials). However, none of these individual correlations survive multiple comparison correction.

## Patterns of dopamine variation are linked to sensitivity to change in travel time, but not decay rate

Next, we sought to examine our secondary behavioral measures of interest, starting with breaking the leaving threshold down into its component parts. In our experiment, reward environments varied on two parameters: travel time between reward patches and decay rate within a patch. To investigate whether the link between the change in patch-leaving threshold and dopamine synthesis and receptor availability was differentially modulated by changes in travel time or decay rate, we measured each individual's sensitivity to these environmental parameters by calculating the change in exit thresholds due to travel time (average threshold for leaving environments with a short travel time – average threshold for leaving environments with a long travel time) and decay rate (average threshold for leaving environments with a shallow decay rate – average threshold for leaving environments with a steep decay rate) separately. The change due to travel time and change due to decay rate scores were normalized across subjects to account for differences in variation between these two measures. These parameters were uncorrelated ($p = 0.314$).

We found that the correlations between patterns of dopamine variability and change in patch-leaving threshold were driven by the travel time and not the depletion rate. To test this, we ran linear regressions for each of the two PCA components that were correlated with total change in patch-leaving threshold (components 1 and 4) including the component score as the dependent variable and the travel time and decay rate change scores as the independent variables. For both components, only patch exit threshold change due to travel time, but not decay rate, was significantly correlated with component score (component 1: threshold change due to travel time $t = 2.579$, $p = 0.0144$, threshold change due to decay rate $t = 0.760$, $p = 0.453$; component 4: threshold change due to travel time $t = 2.116$, $p = 0.0417$, threshold change due to decay rate $t = 0.730$, $p = 0.470$; see Fig. 5). We again ran partial correlation analyses for the individual ROI data, controlling for age and sex, to aid in interpretation and found that threshold change due to travel time was positively correlated with D$_1$ receptor availability in the ACC ($r = 0.306$, $p = 0.0458$) and ventral striatum ($r = 0.388$, $p = 0.0113$) with a trend in the putamen ($p < 0.1$); it

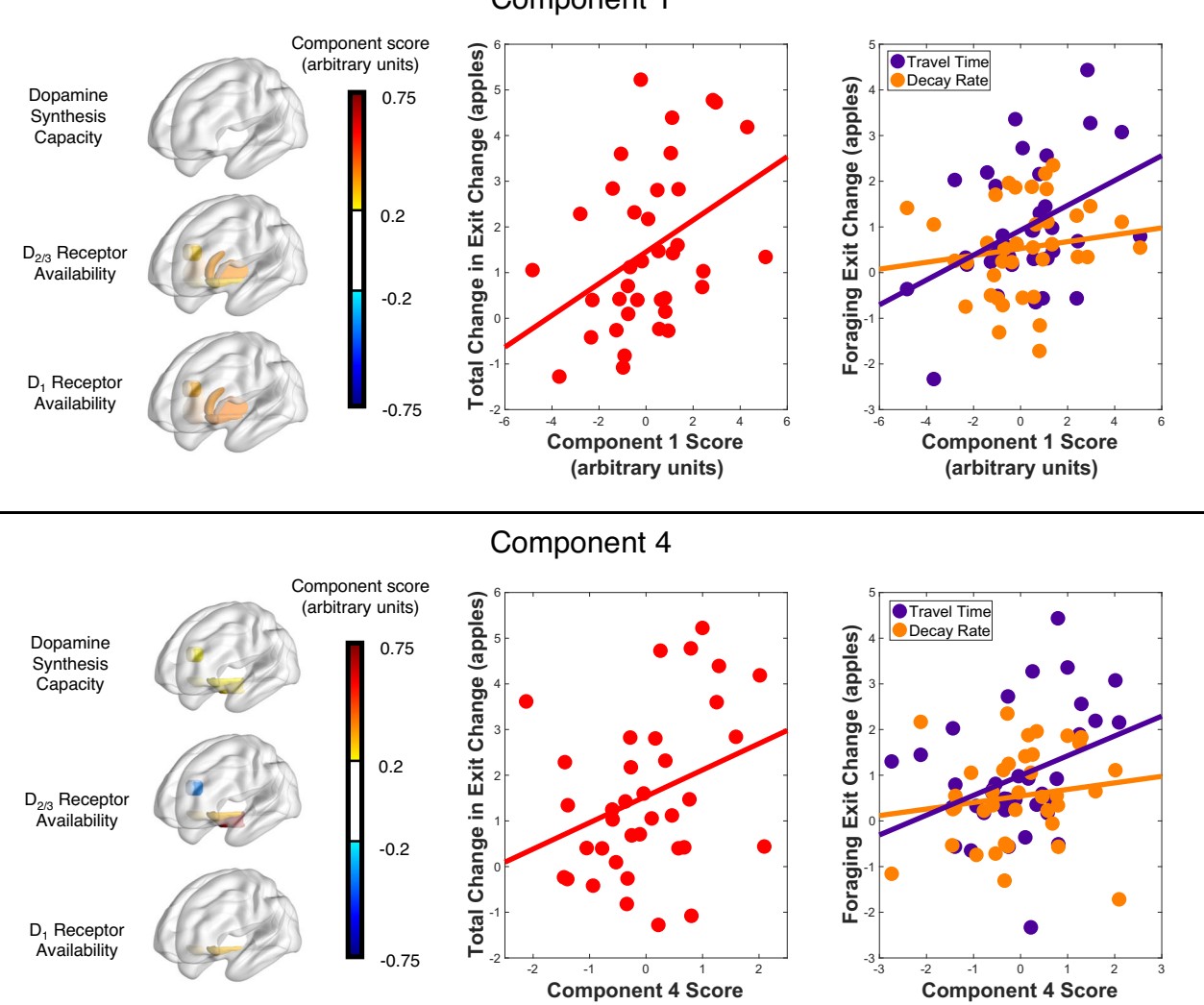

**Fig. 5 | Correlation of dopamine PCA components 1 and 4 with total change in leaving threshold.** Images on the left are the component weightings for PCA components 1 (top) and 4 (bottom). The middle plots (in red) show that individuals who made greater adjustments in their patch-leaving threshold had higher scores for PCA component 1 and component 4 (linear regression model with all four components $F = 3.35$, $p = 0.0213$, two-sided, $n = 37$ individual participants; component 1 $t = 2.699$, $p = 0.011$, component 4 $t = 2.390$, $p = 0.0229$). The right-hand plots show that dopamine PCA components 1 and 4 are positively correlated with change

in travel time and but not with change in decay rate. Component 1 (top): linear regression model $F = 3.78$, $p = 0.0328$, two-sided; threshold change due to travel time $t = 2.579$, $p = 0.0144$, threshold change due to decay rate $t = 0.760$, $p = 0.453$. Component 4 (bottom): linear regression model $F = 2.64$, $p = 0.0861$, two-sided, threshold change due to travel time $t = 2.116$, $p = 0.0417$, threshold change due to decay rate $t = 0.730$, $p = 0.470$. Reported p-values are unadjusted. Source data are provided as a Source Data file.

was also positively correlated with $D_{2/3}$ receptor availability in the caudate nucleus ($r = 0.323$, $p = 0.0482$) with trends in the putamen and ventral striatum ($p < 0.1$; see Supplementary Materials). The individual ROI results do not hold up to multiple comparison correction. There were no correlations between threshold change due to decay rate and any of the ROI measures (all $p > 0.1$). This dissociation provides evidence that these two patterns of dopamine synthesis capacity and receptor availability are implicated in the valuation of time while foraging.

**Change in reaction time is positively correlated with midbrain dopamine synthesis capacity and striatal $D_1$ receptor availability**

Lastly, we analyzed our final secondary behavioral measure, change in response time, to test the computational prediction about dopamine's role in response invigoration. As predicted by experiment design, we confirmed that the average reward rate was highest in the reward environment with the short travel time and shallow depletion

rate and lowest in the reward environment with the long travel time and steep depletion rate (see Fig. 6a). We also confirmed that reaction time decreased in the most rewarding reward environment, suggesting response invigoration (see Fig. 6b). We then ran a single regression model including the two components (1 and 4) found to be implicated in behavioral adjustments to changes in the reward environment as independent variables predicting change in reaction time. Both dopamine patterns of variability were associated with greater change in reaction time between the most and least rewarding environments (component 1 $t = 2.704$, $p = 0.0106$, component 4 $t = 2.406$, $p = 0.0217$; see Fig. 6c, d). Individual ROI analyses revealed positive correlations with change in reaction time and $D_1$ receptor availability in the ventral striatum ($r = 0.378$, $p = 0.0123$) and presynaptic dopamine synthesis capacity in the midbrain ($r = 0.409$, $p = 0.0039$), with trends ($p < 0.1$) between change in reaction time and $D_1$ receptor availability in the ACC and putamen, $D_{2/3}$ receptor availability in the ventral striatum, and presynaptic

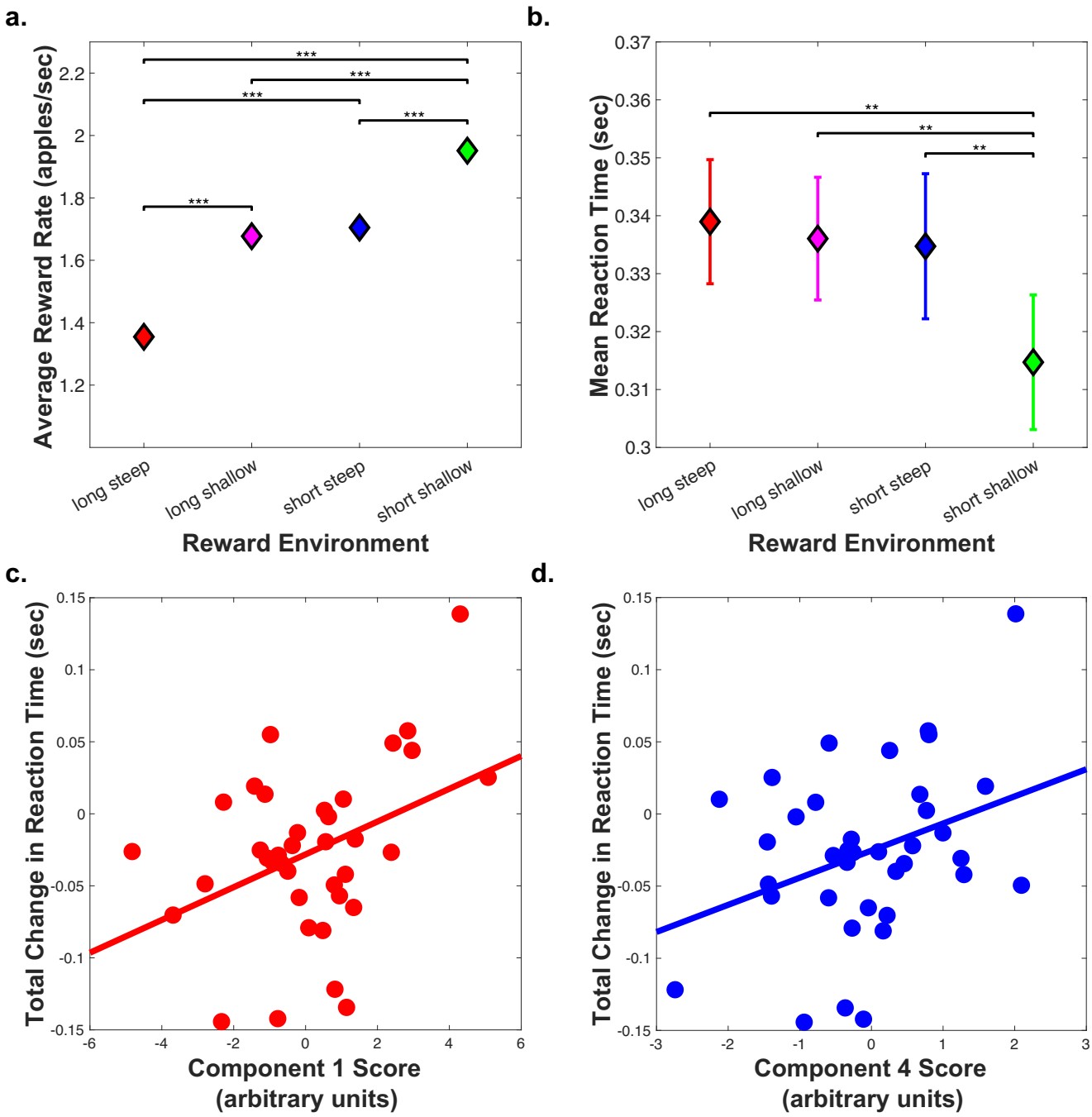

**Fig. 6 | Correlation of dopamine PCA components and change in reaction time.**
**a** Average of the reward rate across subjects in each reward environment ($n = 56$ individual participants). Data are presented as mean +/− standard error of the mean (SEM). Paired t-tests revealed that the average reward rate was lower in the long-steep reward environment compared to the long-shallow ($p = 1.51e−19$), short-steep ($p = 3.47e−21$), and short-shallow ($p = 3.40e−28$) environments. In addition, the average reward rate was higher in the short-shallow reward environment compared the long-shallow ($p = 7.68e−16$) and short-steep ($p = 4.12e−13$) environments.
**b** Average reaction time across subjects in each reward environment. Data are presented as mean +/− SEM. Paired t-tests showed that the reaction time in the short-shallow reward environment is quicker than the long-steep ($p = 0.0022$),

long-shallow ($p = 0.0034$), and short-steep ($p = 0.0045$) environments.
**c** Correlation between total change in reaction time between the short-shallow and long-steep reward environments and dopamine PCA component 1 score ($r = 0.395$, $p = 0.0155$). **d** Correlation between total change in reaction time and dopamine PCA component 4 score ($r = 0.349$, $p = 0.0343$). Linear regression model including both components 1 and 4 scores as the independent variables predicting the dependent variable, change in reaction time, revealed a significant model with $F = 6.57$, $p = 0.00386$. Both components were significantly associated with the change in reaction time: component 1: $t = 2.704$, $p = 0.0106$; component 4: $t = 2.406$, $p = 0.0217$. Results from paired t-tests are indicated on the figure as follows: *$p < 0.05$, **$p < 0.01$, ***$p < 0.001$. Source data are provided as a Source Data file.

dopamine synthesis capacity in the ACC and ventral striatum. The PCA and ROI results suggest the importance of widespread striatal dopamine receptor availability as well as mesolimbic and ACC pre-synaptic dopamine synthesis capacity in response invigoration while foraging.

## Discussion

This study provides valuable insight into the role of dopamine in foraging behavior by measuring multiple facets of the dopamine system in the same individuals who completed a patch foraging task. Through our unique study design and utilization of PCA to combine

PET measures, we found that the degree to which individuals change their foraging behavior based on the parameters of the experimental reward environment is correlated with two patterns of dopamine synthesis capacity and receptor availability throughout the basal ganglia and ACC. First, we showed that the change in reward patch leaving threshold was positively correlated with the availability of dopamine $D_1$ and $D_{2/3}$ receptors throughout the striatum as well as with a combination of increased dopamine synthesis capacity and receptor availability specifically in mesolimbic regions. Next, we found that dopamine function was linked to behavioral changes related to travel time between reward patches and not to depletion rate of rewards within a patch, suggesting a key role of dopamine in calculating the opportunity cost of time. Lastly, we found that these two patterns of high dopamine receptor availability throughout the striatum and high mesolimbic dopamine synthesis capacity were also related to acceleration of response times in the reward environment with the highest average reward rate.

It is possible that in the foraging context striatal presynaptic dopamine synthesis capacity enables fine-tuned adjustments in dopamine release and that greater $D_1$ and $D_{2/3}$ receptor availability enhances sensitivity to changes in local dopamine release in response to changes in the reward environment. Of note, while we are unable to directly measure tonic dopamine in the synapse, preclinical work has shown that presynaptic synthesis capacity is related to dopamine neuron population activity (or average number of spontaneously active dopamine neurons), which is thought to influence tonic dopamine efflux[25–27]. This interpretation builds on computational predictions that tonic striatal dopamine encodes the average reward rate of the environment[10] and supports animal models showing ventral striatum dopamine release tracks average reward rate independently of tonic dopamine firing[29], suggesting an important role of receptor availability in enacting downstream effects. In addition, this finding builds on prior human studies showing administration of an exogenous dopamine $D_2$ receptor agonist[3] alters the reward patch leaving threshold and provides further insight into the role of both $D_1$ and $D_2$ receptors as well as the localization to the mesolimbic dopamine pathway.

We demonstrated that the total change in patch exit threshold between the reward environments with the highest and lowest average reward rate is positively correlated with two separate patterns of dopamine variability. The first pattern is characterized by high $D_1$ and $D_{2/3}$ receptor availability throughout the striatum, which could represent a state of increased reactivity to dopamine release[26]. The positive correlation between dopamine receptor availability and behavioral sensitivity to changes in the reward environment was also reflected in our voxelwise analysis and supports computational and animal work showing that dopamine receptors are important for weighing costs and benefits related to effort costs[11,17,33]. In addition, our results implicating both $D_1$ and $D_{2/3}$ receptors in foraging behavior adds to prior knowledge about the role of $D_2$ receptors on foraging behavior[3] and suggests that both receptor types are important for adjustments in the foraging patch leaving decision threshold. This supports computational models on the dual actions of dopamine receptors for learning and decision making, such that $D_1$ receptors are important for learning the benefits of an action, such as staying in a reward patch, while $D_2$ receptors play a key role in learning about costs, such as the opportunity cost of lost time[11,17]. Therefore, both $D_1$ and $D_2$ receptor-mediated functions are essential for accurately weighing both benefits and costs to adjust behavior based on changes in the environment. In contrast to the prior study of the effect of $D_2$ agonist administration on foraging behavior[3], our study modulated travel time and not just decay rate and also measured both types of dopamine receptors in the same individuals ($D_1$ and $D_{2/3}$), allowing us to identify a key role of dopamine in time valuation. However, in our experiment, the travel time and decay rate parameters were held constant within each reward environment leading to an inability to measure differences in sensitivity to reward environment changes within the rich and poor contexts.

The second pattern of dopamine variability that was correlated with the total change in patch exit threshold includes a positive contribution of presynaptic synthesis capacity in the ACC, ventral striatum, and midbrain, as well as dopamine $D_1$ and $D_{2/3}$ receptor availability in the ventral striatum, all key regions in the reward network. It was particularly interesting that ACC dopamine synthesis capacity was included in this component, given the prior primate and human fMRI studies showing that ACC neural activity encodes information about the reward patch-leaving threshold and the average value of the environment[1,7,15], but changes in patch-leaving threshold cannot be accounted for by neural activity alone[1]. The individual ROI results suggest that ACC $D_1$ receptor availability (included in the first PCA component) may also be important for adjusting the patch-leaving threshold, particularly due to changes in the travel time between patches. It is possible that local dopamine presynaptic release and action at the $D_1$ receptor within the ACC in response to changes in the reward environment may provide a direct mechanism for setting the threshold for leaving a reward patch. In addition, dopamine may modulate the effect of input from other brain regions to the ACC as the striatum and its interactions with the ACC are known to play a role in encoding prediction errors and search costs[7]. Furthermore, this component also includes a positive contribution of $D_{2/3}$ receptor availability in the midbrain, a region where $D_2$ receptors play a key role in the autoregulation of dopamine release throughout the brain[34]. It has been shown that autoreceptors regulate the intrinsic pacemaker activity of dopamine neurons that underlies tonic dopamine levels[35–37], which may be a potential mechanism by which striatal tonic dopamine levels track changes in the average reward rate of the environment. However, the fact that in the individual ROI analyses midbrain $D_{2/3}$ receptor availability was not correlated with changes in patch-leaving threshold suggest that it is unlikely to have an important role outside the context of dopamine function in other regions. In addition, we suggest caution in the generalizability of conclusions regarding relations with this dopamine component since it did not robustly replicate in our independent sample. However, taken together with our voxel-wise results showing that the total change in patch leaving threshold was positively correlated with $D_1$ and $D_{2/3}$ receptor availability throughout the striatum, these findings demonstrate that dopamine synthesis capacity and receptor availability throughout the reward network predicts greater behavioral sensitivity to parameters in the foraging reward environment. Although our PET measures are not direct assays of dopamine release, we speculate that fluctuations in dopamine function in these two identified localized patterns could reflect one potential mechanism by which information about the reward environment could be signaled to neurons in the ACC and reward network that are key for foraging-based decision-making.

Additionally, we found that dopamine is specifically implicated in behavioral adjustments to changes in travel time between reward patches rather than depletion rate within a patch. This supports prior work showing the selective role of striatal dopamine and neural activity (measured by BOLD activation) in encoding reward timing and the opportunity cost of time[29,38,39]. One potential mechanism for dopamine modulating behavioral responsivity to changes in travel time is through negative prediction errors for lapsed time without a reward[30]. A recent study using a prey selection foraging task showed that individuals learned more slowly from negative prediction errors than from positive ones, contributing to an overestimation of environmental quality when rewards depleted[30]. While the depleting patch design of the current study does not allow for trial-wise modeling of decision-making needed for fitting a traditional reinforcement learning model and learning rate, we predict that individuals with greater dopamine synthesis capacity and dopamine receptor availability could

be better equipped to respond to small negative reward prediction errors that occur over unrewarded travel time.

There are limitations to our study that should be kept in mind. Specifically, while PET imaging is the only way to non-invasively directly measure the dopamine synthesis capacity and receptor availability in the living human brain, this methodology has several drawbacks compared to methods used in animal research. These include limited spatial and temporal resolution of neural activity and dopamine system release (compared to individual neural recording, microdialysis, voltammetry, and optical sensors) as well as the inability to isolate and modulate specific regions or neural types (such as can be done with direct neural stimulation and optogenetics). In addition, we were unable to quantify tonic dopamine or receptor availability during the foraging task itself due to the time course of the PET tracers used in this study, requiring 90–180 min to obtain a single image. Similarly, because PET and behavioral measures were not concurrent, and were in some cases separated by months to years, our findings likely reflect a trait-wise association and cannot address the degree to which dopaminergic variables predict behavior at a more precise point in time. Future studies using a combined MRI-PET scanner and a displaceable tracer could potentially be used to investigate how state-wise regional dopamine release corresponds to local changes in foraging decision-making. Lastly, the [18F]-FDOPA tracer quantifies the activity of aromatic l-amino acid decarboxylase (AADC), the enzyme that converts L-dopa to L-dopamine. While in dopamine-rich regions, such as the striatum, it is most likely that variability in this measurement is attributable to dopaminergic terminals, [18F]-FDOPA signal in regions where other AADC-containing cells may be abundant (e.g. the ACC) may not be specific to dopamine[40,41], and interpretations of these data should therefore be cautious.

In conclusion, this study revealed direct correlation between adjustments in foraging behavior and dopamine synthesis capacity and receptor availability in humans. Our results highlight the key role of dopamine receptor availability as well as mesolimbic dopamine function in effecting behavioral changes while foraging such as changes in patch leaving threshold and response invigoration. These two patterns of dopamine variability were selectively associated with changes in the travel time between reward patches and not depletion rate within a patch, suggesting a neural mechanism for encoding the opportunity cost of time. Whether dopamine influences time valuation over and above the context of the specific foraging task studied here must be resolved by future studies. In addition, further work is needed to replicate these findings and tease apart the specific roles of pre- and post-synaptic aspects of dopamine systems in foraging behavior parameters as well as dynamic changes in dopamine release during foraging behavior. Our results provide a potential mechanistic explanation for how ACC neural activity underlying foraging decisions (as measured in prior studies) might be modulated by dopamine to enact a change in patch exit threshold based on the specific parameters of the reward environment.

## Methods
### Study participants
Fifty-seven healthy volunteers aged 21–57 years (mean 35.4 ± 9.9, 29 females) were recruited from the local Washington DC community. Subjects were screened by physician-administered physical and neurological examination, standardized clinical interview (SCID)[42], laboratory tests, and structural MRI read by radiologist to rule out psychiatric, neurological, and major medical illness. The group consisted of 51 individuals of European descent, five African Americans, and one participant of Asian descent. Of the 57 participants, 51 completed the [18F]-FDOPA PET scan (mean age 35.3 ± 9.9, 25 females), 45 completed the [11C]-NNC112 scan (mean age 35.3±9.9, 20 females), and 42 completed the [18F]-Fallypride scan (mean age 35.6 ± 9.7, 17 females). One participant was excluded due to behavior that suggested

they were not following the task instructions (absence of any leave decisions in one of the blocks of the task). All participants provided informed consent and were compensated for their participation. Sex was determined by self-report and was not considered in study design. Study procedures were approved by the NIH Combined Neurosciences Institutional Review Board and Radiation Safety Committee.

### Patch foraging behavioral task
To assess foraging decision-making, participants played a computer-based game in which they foraged for apples (see Fig. 1). This task was developed and validated by Constantino and Daw[2]. Subjects were shown an apple tree and asked to decide whether to stay at the current tree and harvest it for apples or leave and search for a new tree. If they decided to stay, they would receive a certain number of apples, shown below the tree. The apples earned were summed over the entire experiment and converted to monetary payment that was added to the study compensation. After staying at a tree and harvesting it, the number of apples remaining in the tree would decrease according to a depletion rate, similar to how a tree in the wild would gradually run out of apples the longer that an animal ate from it. Subjects then made the stay or leave decision again. If the participants decided to leave, they had to endure a travel time delay until they reached a new tree. There were an infinite number of new trees available to travel to. Participants completed this game in four different reward environments that varied in travel time delay (short 6 s or long 12 s) and reward depletion rate (steep 0.88 or shallow 0.94). Each reward environment, or block, lasted a fixed duration of six and a half minutes. Travel time and depletion rate remained constant throughout the block and the blocks were presented in a random order across participants. If participants did not respond within one second, they received a warning and had to wait two and a half seconds before they were allowed to make another decision. The starting amount of rewards in a tree was drawn from a normal distribution with a mean of 10 and standard deviation of 1, with a maximum value of 15. The reward depletion rates within a patch were drawn from beta distributions such that the steep environments had a mean of 0.88 (alpha = 14.9, beta = 2.03) and the shallow environments had a mean of 0.94 (alpha = 31.6, beta = 1.90).

### Behavioral measures of interest
To assess behavioral sensitivity to the variables in this task, we calculated the total change in threshold for leaving a patch between the reward environment with the highest average reward rate (short travel time and shallow depletion rate) and the one with the lowest average reward rate (long travel time and steep depletion rate). In addition, to assess whether there was a dissociation between changes related to reward timing and magnitude, we determined behavioral sensitivity to travel time and decay rate individually. To measure travel time sensitivity, we took the difference between the average threshold for leaving in the two reward environments with the short travel time and the average threshold for leaving in the two reward environments with the long travel time (short−long). Threshold changes due to depletion rate were calculated as the difference between the average threshold for leaving in the reward environments with a shallow depletion rate and the average threshold for leaving in the reward environments with the steep depletion rate (shallow−steep).

### PET data acquisition
The PET scans were all collected on separate days from the behavioral testing using a GE Advance 3D scanner ([18F]-FDOPA) and Siemens High-Resolution Research Tomograph (HRRT) scanner ([18F]-Fallypride and [11C]-NNC112). Following an eight-minute transmission scan, dynamically binned emission scans were collected for one and a half hours ([18F]-FDOPA and [11C]-NNC112) and four hours ([18F]-Fallypride) after tracer injection. The target tracer doses were 16 mCi for [18F]-FDOPA, 5 mCi for [18F]-Fallypride, and 20 mCi for [11C]-NNC112. For all

scans, caffeine and nicotine were restricted for four hours preceding the scan, and for [18F]-FDOPA only, food was restricted for six hours preceding the scan to reduce competition for transport of the tracer into the brain. Subjects were pretreated with 200 mg carbidopa one hour prior to injection for the [18F]-FDOPA scan to reduce peripheral degradation of the tracer. Subjects also completed T1-weighted magnetic resonance imaging (MRI) scans used for registration and brain segmentation.

### PET data analysis

All PET images underwent attenuation correction, reconstruction, and registration to align all timepoints. We then performed both native-space ROI as well as voxelwise standard-space analyses within the basal ganglia and ACC. For both methods, we used non-invasive, reference input kinetic models for PET parameter modeling, implemented with the PMOD software (http://www.pmod.com/web/). The [18F]-FDOPA specific uptake rate ($K_i$) was calculated with the graphical linearization Gjedde-Patlak method[43,44], while [11C]-NNC112 and [18F]-Fallypride binding potential ($BP_{ND}$, a measure of receptor availability) was calculated with the simplified reference tissue model (SRTM) method[45].

The cerebellum was used as a reference region, delineated on individual native space MRI scans, hand-edited to remove any non-brain voxels, and trimmed to exclude the vermis and lateral and superior parasinus regions using in-house scripts. Specifically, after the MRI was AC-PC aligned, 15 mm was trimmed from the lateral boundaries of the cerebellum, 13 mm in each direction from the midline, and anterior and lateral boundaries were set at y greater than 39 mm and z less than 35 mm. The boundary values were determined from empirical testing and visual inspection.

For ROI analyses, native-space MRI scans were segmented using Freesurfer (https://surfer.nmr.mgh.harvard.edu/) and manual adjustments to generate standard ROIs of the basal ganglia, where dopamine projections and receptors are most abundant: dorsal putamen, dorsal caudate nucleus, ventral striatum, and dopaminergic midbrain (see Fig. 4a). In addition, given the strong a priori knowledge about the role of the ACC in foraging decision-making, an ROI was created as a 5-mm-radius sphere centered on the peak voxel encoding the average value of the foraging environment in the seminal human fMRI study by Kolling et al.[7]. This mask was used to extract values from individual subjects' MNI-space PET images. Outliers were defined as ROI values more than three standard deviations from the mean. [18F]-FDOPA data from the putamen of one subject met this criterion and was excluded.

PET images were corrected for inter-scan motion and realigned using FLIRT (http://fsl.fmirb.ox.ac.uk/fsl/). MRI images were registered to native space mean PET images, and time-activity curves were extracted for each of the ROIs. For voxelwise analyses, each individual's coregistered anatomical MRI image was warped to a common space (MNI) template using ANTS software (http://stnava.github.io/ANTs/) and the resulting transformation matrix was applied to the PET images. Common (MNI) space PET images were smoothed using a three-dimensional Gaussian kernel of 10 mm$^3$ before undergoing modeling in PMOD.

For ROI analyses, linear partial correlations with age and sex as covariates of no interest were run in MATLAB (https://www.mathworks.com/products/matlab.html). Statistical correction for multiple comparisons was done using a false discovery rate (FDR) level of 0.05 (or 5%) as described by Benjamini and Hochberg[46]. Specifically, FDR correction was implemented by hand by ranking the p-values of all correlations tested (n) within each specific tracer from lowest to highest (ranking = i, with the least significant $p$-value having a value of 1). The critical p-value for each correlation was calculated as (1/i)*0.05. The highest p-value that was smaller than the critical value was identified and all correlations with lower p-values were considered significant.

Voxelwise analyses, restricted to a MNI-space basal ganglia mask, were run in FSL (https://www.fmrib.ox.ac.uk/fsl) using randomize (10,000 permutations) for nonparametric permutation testing and threshold-free cluster enhancement (TFCE)[47] with a statistical threshold of $p < 0.05$, family-wise error corrected for multiple comparisons.

### Principal Component Analysis of PET Data and Behavioral Correlations

To take a comprehensive look at dopaminergic function and reduce the dimensionality of our analyses given the five different ROIs for three different PET tracers, we used a PCA to extract patterns of covariance in the dopamine PET data. Only participants who completed all three PET scans were included in this analysis ($n = 37$). PET values were corrected for age and sex and z-normalized before being input into the PCA analysis. Normalization was essential prior to PCA analysis because the parameters of interest for the three PET tracers used vary by multiple orders of magnitude. Only components with eignenvalues greater than one were used in the regression analysis with foraging behavioral measures of interest.

Regression analyses were conducted in MATLAB to assess the relationship between foraging behavioral measures of interest and principal components of dopamine variation. For each behavioral measure of interest (total change in exit threshold, change in exit threshold due to travel time, change in exit threshold due to decay rate, and total change in reaction time), we ran a linear regression analysis using the matlab fitlm function (https://www.mathworks.com/help/stats/fitlm.html) with the behavioral measure as the dependent variable and the dopamine component scores as the independent variables. Lastly, to test for a dissociation between behavioral sensitivity to changes in reward magnitude depletion rate and travel time, the two dopamine PCA components that were correlated with total change in patch-leaving threshold were used as the dependent variables in two separate linear regression models with both behavioral measures included as the independent variables, with a statistical threshold of $p < 0.05$.

### Reporting summary

Further information on research design is available in the Nature Portfolio Reporting Summary linked to this article.

## Data availability

The data generated in this study are provided in the Source Data file accompanying this publication. Additional data are available upon request from the corresponding author (Angela Ianni, ianniam@upmc.edu). Requests must be consistent with individual participant consent and, as appropriate, may be subject to review by the NIH Internal Review Board. Source data are provided with this paper.

## Code availability

Custom Matlab code was used for behavioral and PCA analyses, implemented in MATLAB R2021a (https://www.mathworks.com/products/matlab.html). The code is publicly available on Github (https://github.com/angmirian/Foraging_Dopamine_2023; Zenodo https://doi.org/10.5281/zenodo.8283106). For PET analyses, the following publicly available software was used: MRI segmentation, PET and MRI alignment, and voxelwise statistical analyses using Freesurfer (https://surfer.nmr.mgh.harvard.edu/), FSL (http://fsl.fmirb.ox.ac.uk/fsl/), and ANTS (http://stnava.github.io/ANTs/); PET data compartment modeling using PMOD (http://www.pmod.com/web/).

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

## Acknowledgements

We thank the participants for choosing to spend their time in service of this work. We are also grateful to the National Institutes of Health Positron Emission Tomography Department for their invaluable help and support. This research was supported by the Intramural Research Program of the National Institute of Health (NIH protocols: NCT00004571, NCT00942981, NCT00024622; funding from NIMH project #: ZIAMH002717, awarded to K.B.).

## Author contributions

A.I. wrote the manuscript with input and review from all authors. A.I., D.E., S.C., J.M., P.K., K.B, and T.B. contributed to study design. A.I., D.E., E.B., C.H., M.G., J.M., P.K., and T.B. contributed to data collection and analysis.

## Competing interests

The authors declare no competing interests.
