## [Peer Review File · Nature Communications]

REVIEWER COMMENTS

Reviewer #1 (Remarks to the Author):

Ianni et al. investigate a timely and important issue regarding the link between foraging behavior in humans and individual differences in dopamine neurotransmission effects measured via PET. The paper is very well-written, and the theoretical rationale for the approach is well motivated. A further strong point of the paper is the multi-tracer PET approach, which combines different measures that have previously mostly been studied separately. However, despite my enthusiasm for the research question and the PET approach, I have one major concern, and a few minor comments. My major concern relates to the use of behavioral contrast measures for individual difference analyses without demonstrating the reliability of these measures.

The primary concern I have relates to the psychometric properties of the patch foraging task. PET measures were correlated with behavioral measures obtained on a separate day. By the way, no information is provided on the temporal separation between these sessions, which would need to be included. The validity of this correlational approach critically depends on the reliability (temporal stability) of the behavioral measures calculated from the foraging task data. This is particularly crucial, as all behavioral measures (total change in threshold, change in threshold due to travel time, change in threshold due to decay, change in RT) are contrast measures, i.e., differences in behavioral measures between two experimental conditions. Contrasts between random variables have an increased variance, and this results in lower reliability (e.g. Enkavi et al., 2019). For this correlational approach to be valid, the reliability of the respective behavioral measures would first need to be demonstrated. Note that split-half reliability would of course not be sufficient in this case, due to the temporal separation in the behavioral and PET measurements.

My remaining concerns are more minor in nature. First, in the introduction, mechanisms of foraging were discussed alongside exploration-related findings with respect to dopamine (p.5). This makes sense, but as presently written, the treatment of dopamine and exploration effects misses many recent human papers on this issue. Furthermore, if the literature on exploration is to be included here, it might make sense to more directly compare/contrast this function with foraging as operationalized here.

Second, for completion and transparency, the authors might also wish to state more explicitly the context in which these data were acquired. For example, was the foraging task the only behavioral task measured on the separate testing day, or were other measures obtained?

Reviewer #2 (Remarks to the Author):

This PET study provides evidence that individual differences in dopamine functions are related to foraging behavior. Specifically, they report that striatal D1 and D2 receptor availability impacted the degree to which the patch leave threshold (reward level below which the individual moves to another patch) is adjusted in response to features of the reward environment (combination of reward depletion rate and travel time), and they further demonstrate specific associations between the dopaminergic variables and sensitivity to time travel between foraging patches, as well as between dopaminergic variables and reaction time (vigor). The question itself is important, the rationale for the study is clearly articulated and the results are novel. A strength of the study is its use of 3 measures of dopamine functions (synthesis capacity, D1 and D2 receptor availability) as well as its reasonably large size for a human PET study (which are typically extremely underpowered due to the costs of PET imaging). Given the sample size, the basic result regarding striatal D1 and D2 receptor availability and change in the leave threshold is compelling. However, enthusiasm is dampened by several significant weaknesses.

1) The authors use a PCA approach to break the data down into several components that cut across dopaminergic variables and regions. They justify based on a statement that the PET variables are correlated. The justification is reasonable but has 2 major limitations. First, in the absence of information about the reliability of the solution, it is difficult to know the stability of the solution. Demonstration of stability with bootstrap analysis would improve confidence. 2nd, and more critically, rather than aiding interpretation, the PCA analysis interferes with interpretation of significant effects. For instance, an important analysis examines sensitivity to distance traveled vs. sensitivity to the slope of the decline in reward over time. Because these analyses are only performed for the PCA components, it is harder to draw conclusions about the specific role of D1, D2 or DA synthesis capacity in any given region. The same problem arises for reaction time. We are left without clear information on the relative magnitude of effects of the variables, or whether there is an interaction between synthesis capacity and receptor availability or regions. Supplemental analyses are needed to clarify whether D1, D2, or synthesis variables reach statistical significance when analyzed on their own.

This problem particularly comes into play in terms of the authors' interpretation of the results. For instance, they identify effects in relation to a component that includes the ventral striatum and midbrain dopamine presynaptic synthesis capacity and D1 and D2/3 receptor availability along with high presynaptic synthesis capacity and low D2/3receptor availability in the ACC, but the text seems to emphasize the ACC. For instance, in results they state "as well as the specific pattern of mesolimbic and ACC dopamine function in response invigoration while foraging." Or in the paragraph of the discussion starting at line 409, which focuses just on the role of the ACC. While the authors do enter these discussions by appropriately referring to the region as part of a component, the interpretation is framed in a manner that focuses on a region in isolation, rather than the larger component. This will remain problematic unless the authors demonstrate that the region or dopaminergic variable treated by itself

shows a significant relationship to the effect in question (whether before or after controlling for other measures).

2) There is a problematic equating of dopamine synthesis capacity and “tonic dopamine” in the paper. For instance, in the discussion, line 410 the authors say, “The second pattern of dopamine variability that was correlated with the total change in patch exit threshold includes a positive contribution of tonic dopamine in the ACC, ventral striatum, and midbrain, as well as dopamine D1 and D2/3 receptor availability in the ventral striatum...” No evidence is provided that dopamine synthesis capacity is equivalent to tonic dopamine. Tonic dopamine in the context of the cited computational models refers to dopamine release caused by tonic pacemaker firing. While the amount of dopamine release caused by tonic firing would certainly be deficient in someone with a degeneration of dopamine synthesis capacity, as in Parkinson’s disease, the authors appear to be assuming that higher synthesis capacity leads to higher tonic dopamine release (or extracellular levels) even within a normal range of synthesis capacity. I am not aware of empirical evidence for this assumption. If it exists, the authors should provide it. If it does not, the authors need to constrain or at least qualify their interpretation appropriately. Even if measuring dopamine turnover instead of synthesis capacity, it would still beg the question as to what extent turnover reflected tonic vs phasic activity.

My remaining comments are more minor and should be relatively straightforward to address.

1) Although the rationale for the study is well-described and the introduction is well-written, an issue arises in that the presentation contains an easily missed shift between whether the variables of interest are leave threshold, or the CHANGE in leave threshold across environments. This is a critical distinction, as the primary study variable utilized in the present study is change in leave threshold (a measure of sensitivity to environment) rather than the leave threshold itself. In the introduction, the authors state that experimental evidence indicates that the marginal value theorem closely describes the foraging behavior of wild animals and humans with no mention that there is a whole other source of bias that is not actually being described here and that needs to be controlled for. Drawing the readers’ attention to the fact that their model is about adjustments to the reward environment, rather than baseline individual differences in leave thresholds will make the paper easier to follow and avoid misinterpretation. The authors might also consider providing a supplemental analysis of leave threshold (rather than change in leave threshold). Presumably, they would not see an association, and thus inclusion would improve clarity of their finding.

2) Results Line 301. The authors state “Fluctuations in dopamine function in these two identified localized patterns could reflect one potential mechanism by which information about the reward environment could be signaled to neurons in the ACC and reward network that are key for foraging-based decision-making.” While this statement is true, it is also a leap from the neuroimaging data, which is a static snapshot of dopamine functions, and therefore is not measuring fluctuations. While the authors are clearly aware of the the limitations in temporal resolution of the PET measures, the seeming

leap is not necessary for the presentation of the results. It would be fine in a discussion, if presented with the appropriate qualification that their data did not measure fluctuations (at the point in the paper where the conjecture is made rather than buried in a limitations statement at the end of the paper).

Reviewer #3 (Remarks to the Author):

Ianni and colleagues report on a study of individual differences in dopamine functioning (dopamine synthesis capacity and D1 and D2/3 receptor density) in relation to foraging behavior. The previously validating foraging task was adapted to investigate the propensity to leave a patch as a function of the travel time between patches and the decay rate of a given patch. It has a well-defined optimal leaving threshold (according to Marginal Value Theorem) and has been previously explored both computationally and empirically in individual difference and dopamine drug studies. This study provides the first evidence that individual differences in this task relate to multiple facets of the dopamine system, as measured by multi-modal PET imaging.

Overall, this is a valuable report. The study is well done, the methods are rigorous, and the analyses are well-motivated and reasonable. Importantly, the data offer a precious and rare account of how multiple facets of cortico-striatal dopamine signaling relate to foraging decisions in particular, but also cost-benefit decision-making more broadly.

While the manuscript is well written and thorough, I do have some questions which were left unanswered.

First, while the task design may preclude analyses of trial-wise learning, the rate at which people shift behavior from one context to another could explain a large chunk of the individual differences in the primary outcome measure: the adjustment in the patch leaving threshold across contexts. Given that dopamine is otherwise linked with behavioral and neural plasticity, it would be good to get a sense of how much the correlation between the primary outcome measure (threshold adjustments) and dopamine functioning might be explained by individual differences in learning dynamics rather than - as the Authors infer - individual differences in tonic dopamine signaling of opportunity costs.

To that end, it would be helpful if the Authors showed subject-level data on adjustments in patch leaving thresholds (e.g. a n-trial moving mean of exit points), after changes in context. Is there evidence of systematic, slow (multi-trial) adjustments following a change? Does the rate of adjustment vary systematically with the particular context which subjects just left? Does moving average variance in the

leaving threshold decrease across trials in a given context? If so, do residual leaving thresholds relate to individual differences in dopamine facets after controlling for these learning dynamics?

Second, the primary behavioral measure which the Authors chose to study (adjustments in the leaving threshold across contexts) is reasonable enough, but it also misses out on other potentially informative relationships. Are dopamine measures (or PCs of dopamine measures) related to the mean threshold? What about the proximity to the MVT optimal leaving threshold?

Third, it seems important to ask whether BMI relates to dopamine or behavior. Indeed, BMI has been found to correlate with reward sensitivity and also D2 receptor availability. It may plausibly also relate to radioligand uptake and signal strength. I would be interested to know whether it explains key variance in the individual difference correlations reported here and also to what degree it relates to foraging behavior in general.

Fourth, I noticed that the Authors only reported tests of whether PC 1 and 4 relate to individual differences in the change in RT across contexts. Why did the authors only examine components 1 and 4? In the manuscript, they argue that they restricted their analyses to only these components because only 1 and 4 were found to relate to threshold adjustments. Yet, while there may be some conceptual overlap, mean thresholds and mean RTs are also conceptually dissociable. We cannot be certain that behavioral invigoration is one and the same with MVT-based opportunity costs. As such, it seems reasonable to give the other PCs the same treatment in relation to RTs as in relation to patch leaving thresholds.

Finally, the main focus of the manuscript, as reflected in the title, is that individual differences in dopamine signaling alters how people trade reward against time costs. As such, the manuscript should also point to a growing literature implicating striatal dopamine signaling in biasing sensitivity to the benefits versus costs of action. It may be worthwhile to discuss this finding in context of broader theoretical models of how dopamine plays a role both in the learning and performance of actions relative to their respective benefits and costs (e.g., Collins and Frank (2014; Psychological Review), and Möller and Bogacz (2019; PLoS Comp Bio) and also Westbrook and Frank (2018; Curr Opin in Beh Sci) who have considered how such benefits versus costs mechanisms may also drive vigor in rich environments. In so doing, they could make a more explicit connection between the kinds of reward/travel cost tradeoffs studied in foraging tasks and the broader hypotheses by which dopamine signaling is thought to mediate the tradeoff between benefits and costs more generally.

REVIEWER #1

“Ianni et al. investigate a timely and important issue regarding the link between foraging behavior in humans and individual differences in dopamine neurotransmission effects measured via PET. The paper is very well-written, and the theoretical rationale for the approach is well motivated. A further strong point of the paper is the multi-tracer PET approach, which combines different measures that have previously mostly been studied separately. However, despite my enthusiasm for the research question and the PET approach, I have one major concern, and a few minor comments. My major concern relates to the use of behavioral contrast measures for individual difference analyses without demonstrating the reliability of these measures.”

Comment 1:

“The primary concern I have relates to the psychometric properties of the patch foraging task. PET measures were correlated with behavioral measures obtained on a separate day. By the way, no information is provided on the temporal separation between these sessions, which would need to be included. The validity of this correlational approach critically depends on the reliability (temporal stability) of the behavioral measures calculated from the foraging task data. This is particularly crucial, as all behavioral measures (total change in threshold, change in threshold due to travel time, change in threshold due to decay, change in RT) are contrast measures, i.e., differences in behavioral measures between two experimental conditions. Contrasts between random variables have an increased variance, and this results in lower reliability (e.g. Enkavi et al., 2019). For this correlational approach to be valid, the reliability of the respective behavioral measures would first need to be demonstrated. Note that split-half reliability would of course not be sufficient in this case, due to the temporal separation in the behavioral and PET measurements.”

Response:

Thank you for this query about the psychometric properties of the patch foraging task. We agree that the temporal stability of the measurements collected is an important consideration and are glad for the opportunity to highlight the reliability of our methods. Prior work has shown good test-retest reliability of the foraging task measurements used in our study. In an experiment using a similar foraging paradigm that modulated travel time but not decay rate, 20 healthy adults without neuropsychiatric illness (mean age 61 years, 11 women) completed the patch-foraging task at two visits, approximately a week apart¹. We obtained the data from this experiment and tested the stability of our measures of interest. Both change in exit threshold and change in reaction time were highly correlated across testing dates (change in exit threshold: $r=0.599$, $p=5.22e-3$; change in reaction time: $r=0.774$, $p=6.07e-5$). Furthermore, the mean exit threshold in the individual reward environments was also stable across testing sessions (short: $r=0.589$, $p=6.28e-3$; long: $r=0.599$, $p=5.27e-3$). We have included these data and plots in the Supplementary Materials, highlighted below.

With regards to the PET measures, all three PET tracers have been shown to have good test-retest reliability over weeks to years. Multiple studies investigating ¹⁸F-DOPA have showed good test-retest reliability in striatal regions at 2 months (reliability coefficients 0.681-0.994)²

and up to 2 years (bilateral ICCs ranging from 0.681-0.944)³. Furthermore, studies have shown high reliability of ¹⁸F-Fallypride at 4-6 weeks (intraclass correlation coefficient > 0.8)⁴ and of ¹¹C-NNC at 2-4 weeks (average intraclass correlation coefficient 0.93)⁵.

The participants in our study completed the PET scans within a median of 9 or fewer months of the foraging behavioral task ([¹⁸F]-FDOPA 9.0 months, [¹⁸F]-fallypride 9.0 months, [¹¹C]-NNC112 5.5 months), which is within a time period that we would not anticipate a significant decline in dopamine synthesis capacity or receptor availability in healthy adults. We now note this information in the main text of the manuscript.

However, because we did include subjects with substantial interscan intervals, we cannot rule out the possibility that our results include interval-related noise. Although, importantly, we would hypothesize that such noise would be more likely to obscure true findings than generate false ones, we now acknowledge this limitation in the Discussion section of the manuscript. Finally, we have conducted additional post-hoc partial correlation ROI analyses including the interval between behavioral task and PET scan as an additional covariate of no interest, which we report in Supplementary Materials. The results from these analyses were unchanged from the original results.

In the manuscript:

We added the following to the main text (p. 10, first paragraph):

“The foraging task was completed during a behavioral testing session that included one other probabilistic decision-making task, either before or after an MRI scan for a different decision-making task. The order of the behavioral tasks was randomly counterbalanced across participants and the order of behavioral testing session and MRI scan was determined by logistical constraints. The behavioral task was collected on a separate day from the PET scans with a median of 9.0 months between the behavioral task and the [¹⁸F]-FDOPA and [¹⁸F]-Fallypride scans and 5.5 months between the behavioral task and the [¹¹C]-NNC112 scan. Sensitivity analyses controlling for time between behavioral task and PET scan are included in the Supplementary Materials. The results from these analyses were unchanged from the original results.”

Furthermore, in the Discussion (p. 25, 1st paragraph):

“Similarly, because PET and behavioral measures were not concurrent, we cannot rule out the possibility that our results suffer from additional noise due to intervals between measurements, although partial correlation analyses including interval as a covariate are reassuring in this regard (see Supplementary Materials).”

In addition, we added the following to the Supplementary Materials (p. 9, second paragraph):

“Temporal Stability of Results

There was a range of temporal separation between the behavioral task and PET scans; however, the median separation time was 9.00 months for [¹⁸F]-FDOPA (interquartile range 47.5 months), 9.00 months for [¹⁸F]-Fallypride (interquartile range 22.5 months), and 5.50 months for [¹¹C]-NNC112 (interquartile range 26.0 months).

Sensitivity analyses controlling for time between behavioral task and PET scan are included in the Supplementary Materials.

Test-Retest Reliability of Foraging Behavioral Measures

In an experiment using a similar foraging paradigm that modulated travel time but not decay rate, 20 healthy adults without neuropsychiatric illness (mean age 61 years, 11 women) completed the patch-foraging task at two visits, approximately a week apart¹⁸. We obtained the data from this experiment and tested the stability of our measures of interest. Both change in exit threshold and change in reaction time were consistent across testing dates (change in exit threshold: $r=0.599$, $p=5.22e-3$; change in reaction time: $r=0.774$, $p=6.07e-5$; plots of these data are included below). Furthermore, the mean exit threshold in the individual reward environments were also stable across testing sessions (short: $r=0.589$, $p=6.28e-3$; long: $r=0.599$, $p=5.27e-3$).

ROI Results Controlling for Time Between Behavioral Task and PET Scan

To investigate whether the ROI results were affected by variable amounts of time elapsed between the PET scans and the foraging task, we performed additional partial correlation analyses including time as an additional covariate of no interest. Adding this covariate did not change our results. We still found a positive correlation between total change in leaving threshold and the same PET measures: D₁ receptor availability positive correlation in the ventral striatum ($r=0.399$, $p=0.0088$) with trends in the ACC ($r=0.295$, $p=0.0582$) and putamen ($r=0.299$, $p=0.0545$); D_{2/3} receptor availability positive trends in the putamen ($r=0.295$, $p=0.0766$), caudate nucleus ($r=0.302$, $p=0.0693$), and ventral striatum ($r=0.310$, $p=0.0619$); presynaptic dopamine synthesis capacity trend in the ACC ($r=0.246$, $p=0.0954$). We also saw the same results for change in reaction time: D₁ receptor availability positive correlation in the ventral striatum ($r=0.476$, $p=0.0014$), caudate nucleus ($r=0.413$, $p=0.0066$), putamen ($r=0.317$, $p=0.0411$), and ACC ($r=0.327$, $p=0.0346$); D_{2/3} receptor availability positive correlation in the ventral striatum ($r=0.335$, $p=0.0427$) with trend in the putamen ($r=0.298$,

p=0.0732); presynaptic dopamine synthesis capacity positive correlation in the midbrain (r=0.408, p=0.0045) with trends in the ACC (r=0.252, p=0.0881) and ventral striatum (r=0.255, p=0.0831)."

Comment 2:

"My remaining concerns are more minor in nature. First, in the introduction, mechanisms of foraging were discussed alongside exploration-related findings with respect to dopamine (p.5). This makes sense, but as presently written, the treatment of dopamine and exploration effects misses many recent human papers on this issue. Furthermore, if the literature on exploration is to be included here, it might make sense to more directly compare/contrast this function with foraging as operationalized here."

Response:

We appreciate the reviewer bringing up this concern and suggestion. To this point, we added relevant results from human studies of dopamine's role in explore-exploit behavior including Chakroun et al 2020⁶ and Frank et al 2009⁷. Please see the additions to the manuscript highlighted below.

In the manuscript:

Page 3, paragraph 1:

"In contrast to explore-exploit decision-making paradigms where the decision is to repeat (exploit) a familiar action or explore a new one, the important choice when foraging is whether to engage with the current environment or leave and search elsewhere."

Page 7, paragraph 1:

"In a continuous-space explore-exploit task, genetic variation in D₁ and D₂ receptor expression was associated with complimentary roles in adjusting response times to maximize rewards, with D₁-receptors implicated in speeding up and D₂-receptors associated with slowing down responses²³. However, a more classic explore-exploit multi-arm bandit task did not find any changes in exploration or exploitation behavior with exogenous administration of a D₂-receptor antagonist²⁴."

Page 7, paragraph 1:

"Finally, both explore-exploit and foraging decision making involve balancing the benefits of exploiting or sticking with a familiar option (e.g. staying in the current reward patch) with exploring alternative options (e.g. leaving for a new reward patch)."

Comment 3:

"Second, for completion and transparency, the authors might also wish to state more explicitly the context in which these data were acquired. For example, was the foraging task the only behavioral task measured on the separate testing day, or were other measures obtained?"

Response:

The foraging behavioral task was collected in a behavioral testing session that included one other probabilistic decision-making task⁸, which was randomly counter balanced across participants. The behavioral testing was completed either before or after an MRI scan for a different credit-assignment decision-making task⁹, with the order of behavioral testing session and MRI scan determined by logistical constraints such as availability of testing room, scanner time, and participant schedule. We have added this information to the manuscript.

In the manuscript:

Included the following (p. 10, first paragraph):

“The foraging task was completed during a behavioral testing session that included one other probabilistic decision-making task, either before or after an MRI scan for a different decision-making task. The order of the behavioral tasks was randomly counterbalanced across participants and the order of behavioral testing session and MRI scan was determined by logistical constraints.”

REVIEWER #2

“This PET study provides evidence that individual differences in dopamine functions are related to foraging behavior. Specifically, they report that striatal D1 and D2 receptor availability impacted the degree to which the patch leave threshold (reward level below which the individual moves to another patch) is adjusted in response to features of the reward environment (combination of reward depletion rate and travel time), and they further demonstrate specific associations between the dopaminergic variables and sensitivity to time travel between foraging patches, as well as between dopaminergic variables and reaction time (vigor). The question itself is important, the rationale for the study is clearly articulated and the results are novel. A strength of the study is its use of 3 measures of dopamine functions (synthesis capacity, D1 and D2 receptor availability) as well as its reasonably large size for a human PET study (which are typically extremely underpowered due to the costs of PET imaging). Given the sample size, the basic result regarding striatal D1 and D2 receptor availability and change in the leave threshold is compelling. However, enthusiasm is dampened by several significant weaknesses.”

Comment 1a:

“1) The authors use a PCA approach to break the data down into several components that cut across dopaminergic variables and regions. They justify based on a statement that the PET variables are correlated. The justification is reasonable but has 2 major limitations. First, in the absence of information about the reliability of the solution, it is difficult to know the stability of the solution. Demonstration of stability with bootstrap analysis would improve confidence.”

Response:

The reviewer brings up an excellent question about the stability of the PCA results, although the PCA results are also supported by the individual tracer voxelwise and ROI results. To address this question, we used two approaches to assess the stability of the PCA solution. First, we examined an independent sample of 26 individuals who had also completed all three PET scans (age 18-49 years, 12 females). Second, we used a bootstrapping sampling approach (with 1000 iterations) to draw random samples of participants with group size 37 (to match the original PCA analysis group) and calculated the confidence interval for the correlation coefficients for each component. We found that component 1 was stable in both the replication and bootstrapping analyses (correlation between component 1 coefficient in original sample and replication sample: $r=0.702$, $p=0.0051$; 95% confidence interval for the correlation coefficient of component 1 coefficient was 0.674-0.708). The other components had lower correlations between samples, although the correlation with component 2 was significant at the $p<0.05$ threshold (component 2: replication sample $r=-0.5592$, $p=0.0376$, bootstrapping 95% CI for the absolute value of the correlation coefficient=0.518-0.554; component 3: replication sample $r=0.169$, $p=0.5626$, bootstrapping 95% CI for the absolute value of the correlation coefficient=0.518-0.554; component 4: replication sample $r=0.0669$, $p=0.820$, bootstrapping 95% CI for the absolute value of the correlation coefficient=0.3749-0.4053). The confidence intervals were similar when using a larger sample for the bootstrapping sampling of

50 rather than 37 subjects (component 1: 0.672-0.707, component 2: 0.507-0.544, component 3: 0.506-0.539, component 4: 0.394-0.425).

In summary, we have shown that both component 1 (globally high dopamine D₁ and D_{2/3} receptor availability) and component 2 (globally discordant D₁ and D_{2/3} receptor availability) represent replicable patterns of variance across individuals. Components 3 and 4, which represent more inter-regional variability in dopamine patterns are less consistent across groups of participants. Therefore, the results for those components should be interpreted in conjunction with the individual tracer voxelwise and ROI analyses to provide additional support of the contribution of regional dopamine patterns. The individual ROI correlations with behavioral measures are now included in the Supplementary Materials.

In the manuscript:

We included the following in the Supplementary Materials (p. 4, first paragraph):

“Reliability of PET PCA Results

To assess the stability of the PCA solution, we used two approaches. First, we examined an independent sample of 26 individuals who had also completed all three PET scans (age 18-49 years, 12 females). Second, we used a bootstrapping sampling approach (with 1000 iterations) of the combined sample of our original subjects and the additional 26 subjects (total = 63 subjects). We drew random samples of participants with group size 37 (to match the original PCA analysis group) and calculated the confidence interval for the correlation coefficients for each component. We found that component 1 was stable in both the replication and bootstrapping analyses (correlation between component 1 coefficient in original sample and replication sample: $r=0.702$, $p=0.0051$; 95% confidence interval for the correlation coefficient of component 1 coefficient was 0.674-0.708). The other components had lower correlations between samples, although the correlation with component 2 was significant at the $p<0.05$ threshold (component 2: replication sample $r=-0.5592$, $p=0.0376$, bootstrapping 95% CI for the absolute value of the correlation coefficient=0.518-0.554; component 3: replication sample $r=0.169$, $p=0.5626$, bootstrapping 95% CI for the absolute value of the correlation coefficient=0.518-0.554; component 4: replication sample $r=0.0669$, $p=0.820$, bootstrapping 95% CI for the absolute value of the correlation coefficient=0.3749-0.4053).”

Comment 1b:

“2nd, and more critically, rather than aiding interpretation, the PCA analysis interferes with interpretation of significant effects. For instance, an important analysis examines sensitivity to distance traveled vs. sensitivity to the slope of the decline in reward over time. Because these analyses are only performed for the PCA components, it is harder to draw conclusions about the specific role of D1, D2 or DA synthesis capacity in any given region. The same problem arises for reaction time. We are left without clear information on the relative magnitude of effects of the variables, or whether there is an interaction between synthesis capacity and receptor availability or regions. Supplemental analyses are needed to clarify whether D1, D2, or synthesis variables reach statistical significance when analyzed on their own.”

Response:

We appreciate the reviewer's comment about these additional analyses that would help aid in interpretation of our results. To this end, we ran partial correlation analyses for each individual PET ROI value and the foraging behavioral measures of interest including total change in reaction time, change in leaving threshold due to travel time, and change in leaving threshold due to decay rate. We have included these results in the Supplementary Materials.

In the manuscript:

Added to the Supplementary Materials (starting at the top of p. 5):

“Individual ROI Correlations with Total Change in Patch-Leaving Threshold

For the total change in patch-leaving threshold between the most and least rewarding environments, we found a positive correlation with D₁ receptor availability in the ventral striatum ($r=0.378$, $p=0.0123$), and trends in the ACC ($r=0.286$, $p=0.0626$) and putamen ($r=0.274$, $p=0.0752$). In addition, there were positive trends between total change in patch-leaving threshold and D_{2/3} receptor availability in the putamen ($r=0.279$, $p=0.0896$), caudate nucleus ($r=0.294$, $p=0.0732$), and ventral striatum ($r=0.305$, $p=0.0623$). Lastly, there was a positive trend between total change in patch-leaving threshold and dopamine presynaptic synthesis capacity in the ACC ($r=0.251$, $p=0.0857$). No regions were significant after correcting for multiple comparisons across all 14 ROIs tested.

Individual ROI and PCA Correlations with Total Change in Reaction Time

With regards to the total change in reaction time between the most and least rewarding environments, we found a positive correlation with D₁ receptor availability in the ventral striatum ($r=0.378$, $p=0.0123$) and trends in the ACC ($r=0.286$, $p=0.0626$) and putamen ($r=0.274$, $p=0.0752$). In addition, we found a positive trend with D_{2/3} receptor availability in the ventral striatum ($r=0.300$, $p=0.0676$). Lastly, we found a positive correlation between change in reaction time and presynaptic dopamine synthesis capacity in the midbrain ($r=0.409$, $p=0.0039$) with trends in the ACC ($r=0.267$, $p=0.0663$) and ventral striatum ($r=0.261$, $p=0.0732$). Again, no regions were significant after correcting for multiple comparisons.

Individual ROI Correlations with Change in Leaving Threshold due to Travel Time and Decay Rate

Decomposing the change in leaving threshold down into the effects of travel time and decay rate, we found that change in threshold due to travel time was positively correlated with D₁ receptor availability in the ACC ($r=0.306$, $p=0.0458$) and ventral striatum ($r=0.383$, $p=0.0113$) with a trend in the putamen ($r=0.285$, $p=0.0641$). Change in leaving threshold due to travel time was also positively correlated with D_{2/3} receptor availability in the caudate nucleus ($r=0.323$, $p=0.0482$) with trends in the putamen ($r=0.293$, $p=0.0745$) and ventral striatum ($r=0.277$, $p=0.0920$). Lastly, there was a positive trend between change in leaving threshold due to travel time and presynaptic dopamine synthesis capacity in the ACC ($r=0.242$, $p=0.0972$) and midbrain ($r=0.245$, $p=0.0931$). None of these correlations were significant after multiple

comparison correction. There were no significant correlations or trends between change in leaving threshold due to decay rate and any of the PET measures (all $p > 0.1$)."

Comment 1c:

"This problem particularly comes into play in terms of the authors' interpretation of the results. For instance, they identify effects in relation to a component that includes the ventral striatum and midbrain dopamine presynaptic synthesis capacity and D1 and D2/3 receptor availability along with high presynaptic synthesis capacity and low D2/3 receptor availability in the ACC, but the text seems to emphasize the ACC. For instance, in results they state "as well as the specific pattern of mesolimbic and ACC dopamine function in response invigoration while foraging." Or in the paragraph of the discussion starting at line 409, which focuses just on the role of the ACC. While the authors do enter these discussions by appropriately referring to the region as part of a component, the interpretation is framed in a manner that focuses on a region in isolation, rather than the larger component. This will remain problematic unless the authors demonstrate that the region or dopaminergic variable treated by itself shows a significant relationship to the effect in question (whether before or after controlling for other measures)."

Response:

Thank you for highlighting this problematic wording in the manuscript. We have taken the opportunity to adjust the text to provide clarity for the reader. We have also added the individual ROI results to aid in interpretation.

In the manuscript:

We changed the wording in the last paragraph of the Results section (p. 19, first paragraph):

"The PCA and ROI results suggest the importance of widespread striatal dopamine receptor availability as well as mesolimbic and ACC presynaptic dopamine synthesis capacity in response invigoration while foraging."

In addition, we modified the paragraph in the discussion about this component and the ACC as follows (p. 22, second paragraph):

"It was particularly interesting that ACC dopamine synthesis capacity was included in this component, given the prior primate and human fMRI studies showing that ACC neural activity has been shown to encode information about the reward patch-leaving threshold and the average value of the environment^{1,7,15}, but changes in patch-leaving threshold cannot be accounted for by neural activity alone¹. The individual ROI results suggest that ACC D₁ receptor availability (included in the first PCA component) may also be important for adjusting the patch-leaving threshold, particularly due to changes in the travel time between patches. It is possible that local dopamine presynaptic release and action at the D₁ receptor within the ACC in response to changes in the reward environment may provide a direct mechanism for setting the threshold for leaving a reward patch. In addition, dopamine may modulate the effect of input from other brain regions to the ACC as the striatum and its interactions with the ACC are known to play a role in encoding prediction errors and search costs⁷."

Comment 2:

“2) There is a problematic equating of dopamine synthesis capacity and “tonic dopamine” in the paper. For instance, in the discussion, line 410 the authors say, “The second pattern of dopamine variability that was correlated with the total change in patch exit threshold includes a positive contribution of tonic dopamine in the ACC, ventral striatum, and midbrain, as well as dopamine D1 and D2/3 receptor availability in the ventral striatum...” No evidence is provided that dopamine synthesis capacity is equivalent to tonic dopamine. Tonic dopamine in the context of the cited computational models refers to dopamine release caused by tonic pacemaker firing. While the amount of dopamine release caused by tonic firing would certainly be deficient in someone with a degeneration of dopamine synthesis capacity, as in Parkinson’s disease, the authors appear to be assuming that higher synthesis capacity leads to higher tonic dopamine release (or extracellular levels) even within a normal range of synthesis capacity. I am not aware of empirical evidence for this assumption. If it exists, the authors should provide it. If it does not, the authors need to constrain or at least qualify their interpretation appropriately. Even if measuring dopamine turnover instead of synthesis capacity, it would still beg the question as to what extent turnover reflected tonic vs phasic activity.”

Response:

We appreciate this important point. Per the reviewer’s suggestion, we now include literature evidence for associations between tonic dopamine release and synthesis capacity. The most recent data addressing this point come from preclinical experiments of methylazoxymethanol acetate (MAM) treatment in rats, a model thought to recapitulate dopaminergic deficits seen in schizophrenia¹⁰. In these experiments, MAM treatment specifically elevated dopaminergic neuron population activity (a metric reflecting the proportion of spontaneously active dopamine neurons), which corresponded to concurrent increase in presynaptic dopamine synthesis capacity measured with [³H]-DOPA autoradiography. We now cite this important work in the manuscript as well as others linking tonic dopamine and presynaptic synthesis capacity^{11,12}. Furthermore, taking the reviewer’s comment to heart, we additionally have more carefully constrained language around ‘tonic dopamine’ interpretations and references throughout the manuscript.

In the manuscript:

In the introduction (p. 8):

“Preclinical work in rodent models has suggested that presynaptic synthesis capacity may be related to constitutive dopamine neuron population activity (or average number of spontaneously active dopamine neurons), which is thought to influence tonic dopamine efflux²⁵⁻²⁷.”

In the discussion (p. 20, second paragraph):

“Of note, while we are unable to directly measure tonic dopamine in the synapse, preclinical work has shown that presynaptic synthesis capacity is related to dopamine neuron population activity (or average number of spontaneously active dopamine neurons), which is thought to influence tonic dopamine efflux²⁵⁻²⁷.”

Comment 3:

“My remaining comments are more minor and should be relatively straightforward to address.

1) Although the rationale for the study is well-described and the introduction is well-written, an issue arises in that the presentation contains an easily missed shift between whether the variables of interest are leave threshold, or the CHANGE in leave threshold across environments. This is a critical distinction, as the primary study variable utilized in the present study is change in leave threshold (a measure of sensitivity to environment) rather than the leave threshold itself. In the introduction, the authors state that experimental evidence indicates that the marginal value theorem closely describes the foraging behavior of wild animals and humans with no mention that there is a whole other source of bias that is not actually being described here and that needs to be controlled for. Drawing the readers’ attention to the fact that their model is about adjustments to the reward environment, rather than baseline individual differences in leave thresholds will make the paper easier to follow and avoid misinterpretation. The authors might also consider providing a supplemental analysis of leave threshold (rather than change in leave threshold). Presumably, they would not see an association, and thus inclusion would improve clarity of their finding.”

Response:

The reviewer brings up an excellent point and we agree that emphasizing this concept in the introduction would be beneficial for the reader. We have added clarification to the paragraph introducing the marginal value theorem, as noted below. In addition, we have added the mean patch-leaving threshold results to the Supplementary Material to aid in interpretation.

In the manuscript:

Introduction (p. 4, end of first paragraph):

“Of note, experimental data from humans, nonhuman primates, and other animals has shown a deviation from the MVT such that there is a consistent bias to stay in reward patches longer than optimal^{1,2,7-9}. This could reflect factors not accounted for in the MVT such as preference for immediate over delayed rewards, risk of predation during travel between reward patches, activities that occur simultaneously during foraging (e.g. parental care, searing for mates), varied nutritional states (e.g. hungry vs. satiated)^{8,9}. However, past studies have shown that measuring the relative change in patch leaving threshold between reward environments controls for individuals’ bias to stay and more closely reflects optimal behavior modeled with the MVT^{3,7}.”

Supplementary Materials (starting at the bottom of p. 4):

“PCA Component Correlations with Mean Patch-Leaving Thresholds

To aid in interpretation of our threshold change results, we also ran linear regressions with mean patch-leaving threshold (across all four reward environments as well as each individually) as the dependent variable and the four dopamine PET PCA component scores as the independent variables. The mean patch-leaving

threshold across all reward environments was not related to the dopamine PCA component scores (complete model $p=0.719$, individual component score p -values >0.3). When looking at the mean leaving thresholds for the individual reward environments, we found that the mean threshold for reward environment with the higher average reward rate (short travel time and shallow decay rate) was positively correlated with component 1 score (t -stat=2.136, $p=0.0404$) although the complete regression model was not significant ($p=0.141$). There were no significant correlations with the PCA component scores and the mean leaving thresholds for any of the other reward environments (all $p>0.1$)."

Individual ROI Correlations with Mean Patch-Leaving Threshold

We ran linear partial correlations between each PET ROI value and our behavioral measures of interest, controlling for age and gender. There are no significant correlations between mean patch-leaving threshold across all environments and the individual PET ROI values (smallest p -value is 0.1794). When looking at the individual reward environment patch-leaving thresholds, there are significant positive correlations between the leaving-threshold for the short-shallow reward environment and $D_{2/3}$ receptor binding potential in the caudate nucleus ($r=0.3141$, $p=0.0455$), ventral striatum ($r=0.3216$, $p=0.0403$), and a trend in the putamen ($r=0.2958$, $p=0.0604$). There is also a trend towards a positive correlation with D_1 receptor binding potential in the ventral striatum ($r=0.2743$, $p=0.0751$). Dopamine presynaptic synthesis capacity is not correlated with any of the individual environment leaving thresholds or with D_1 or $D_{2/3}$ receptor binding potential and the leaving threshold in any of the other reward environments (all $p>0.1$)."

Comment 4:

"2) Results Line 301. The authors state "Fluctuations in dopamine function in these two identified localized patterns could reflect one potential mechanism by which information about the reward environment could be signaled to neurons in the ACC and reward network that are key for foraging-based decision-making." While this statement is true, it is also a leap from the neuroimaging data, which is a static snapshot of dopamine functions, and therefore is not measuring fluctuations. While the authors are clearly aware of the limitations in temporal resolution of the PET measures, the seeming leap is not necessary for the presentation of the results. It would be fine in a discussion, if presented with the appropriate qualification that their data did not measure fluctuations (at the point in the paper where the conjecture is made rather than buried in a limitations statement at the end of the paper)."

Response:

We agree with the reviewer's point that the highlighted statement is more appropriate for the discussion section with the noted qualifications about limitations of the methods. We've adjusted the text accordingly.

In the manuscript:

Removed the sentence highlighted from the results section and added the following to the discussion (bottom of p. 23):

“Although our PET measures are not direct assays of dopamine release, we speculate that fluctuations in dopamine function in these two identified localized patterns could reflect one potential mechanism by which information about the reward environment could be signaled to neurons in the ACC and reward network that are key for foraging-based decision-making.”

REVIEWER #3

"Ianni and colleagues report on a study of individual differences in dopamine functioning (dopamine synthesis capacity and D1 and D2/3 receptor density) in relation to foraging behavior. The previously validating foraging task was adapted to investigate the propensity to leave a patch as a function of the travel time between patches and the decay rate of a given patch. It has a well-defined optimal leaving threshold (according to Marginal Value Theorem) and has been previously explored both computationally and empirically in individual difference and dopamine drug studies. This study provides the first evidence that individual differences in this task relate to multiple facets of the dopamine system, as measured by multi-modal PET imaging.

Overall, this is a valuable report. The study is well done, the methods are rigorous, and the analyses are well-motivated and reasonable. Importantly, the data offer a precious and rare account of how multiple facets of cortico-striatal dopamine signaling relate to foraging decisions in particular, but also cost-benefit decision-making more broadly.

While the manuscript is well written and thorough, I do have some questions which were left unanswered."

Comment 1a:

"First, while the task design may preclude analyses of trial-wise learning, the rate at which people shift behavior from one context to another could explain a large chunk of the individual differences in the primary outcome measure: the adjustment in the patch leaving threshold across contexts. Given that dopamine is otherwise linked with behavioral and neural plasticity, it would be good to get a sense of how much the correlation between the primary outcome measure (threshold adjustments) and dopamine functioning might be explained by individual differences in learning dynamics rather than - as the Authors infer - individual differences in tonic dopamine signaling of opportunity costs.

To that end, it would be helpful if the Authors showed subject-level data on adjustments in patch leaving thresholds (e.g. a n-trial moving mean of exit points), after changes in context. Is there evidence of systematic, slow (multi-trial) adjustments following a change?"

Response:

We are grateful for the reviewer bringing up this interesting point. We plotted the subject-level data on adjustments in patch leaving threshold as an n-trial moving mean of exit threshold for each reward environment and the leaving threshold appears to be stable (see plot below).

Average exit threshold within subject within reward environment

Running average reward patch exit thresholds within each reward patch.

Comment 1b:

“Does the rate of adjustment vary systematically with the particular context which subjects just left?”

Response:

There do not appear to be any systematic adjustments in the patch leaving threshold based on the previous reward environment either, both when plotted according to the current block and colored by previous block, or when plotted based on the previous block (see plots below).

Average exit threshold within subject colored by PRIOR block

Running average reward patch exit thresholds within each reward patch colored by prior block (note first-block indicates that the plotted block is the first block of the experiment)

Running average reward patch exit thresholds grouped by previous reward environment, colored by subject ID (note first-block indicates that the plotted block is the first block of the experiment)

Comment 1c:

“Does moving average variance in the leaving threshold decrease across trials in a given context?”

Response:

To address this question, we calculated the average leaving threshold and standard deviation across all subjects for the first five exit decisions, which appears to plateau off after the first 2-3 exit decisions (see plot below). Given that some subjects had limited exit decisions specific reward blocks, we were limited to only looking at the first five exit decisions. Likewise, we were unable to calculate within subject variance in leaving threshold across trials because many subjects only have a few exit decisions in each block. However, based on the plots of average exit threshold within subject shown above, it appears that individuals’ leaving threshold becomes stable after a few exit decisions.

The average reward patch exit threshold and standard deviation across subjects at each exit decision within block.

Furthermore, we found that the first exit threshold and final average exit threshold for each subject are highly correlated, supporting the stability of exit thresholds within subjects (see plot below).

Within-subject correlation between exit threshold at the beginning of the block and the final average exit threshold for each block

Comment 1d:

“If so, do residual leaving thresholds relate to individual differences in dopamine facets after controlling for these learning dynamics?”

Response:

To assess for individual differences in learning dynamics we calculated the slope of the average exit threshold over exit decisions for the first five exit decisions for each subject in each reward environment. We then calculated the difference in the slopes between the environments with the highest (short-shallow) and lowest (long-steep) average reward rates. We ran partial correlations between our primary behavioral measure (total change in exit threshold between the short-shallow and long-steep reward environments) and the dopamine component scores, controlling for the difference in slopes between these two reward environments. Both component 1 and 4 remained significantly correlated with the total change in exit threshold even after controlling for the difference in slopes (component 1 $p=3.01e-2$, component 4 $p=2.22e-2$).

Finally, to reduce potential confounds of initial learning-related adjustments, we recalculated the average exit threshold for each subject excluding their first 3 exit decisions within each block. The average leaving thresholds with and without the first three exit decisions were highly correlated (short-shallow environment $r=0.970$, $p=6.78e-35$; short-steep environment $r=0.977$, $p=7.17e-38$; long-shallow environment $r=0.940$, $p=2.19e-26$; long-steep environment $r=0.969$, $p=1.49e-24$). We recalculated the total change in leaving threshold using these filtered exit decisions and including these in the linear regression model with PCA components and our results were unchanged (component 1 $p=1.04e-2$, component 4 $p=6.44e-3$).

In the manuscript:

We have added the following to the Supplementary Materials (starting at the top of p. 2).

“Relative stability of average leaving threshold within reward environment

The running average of the patch leaving threshold plotted over exit decisions showed overall stability of the average and no apparent effects of block (see Supplementary Figure 2) or previous block (see Supplementary Figure 2). Exit thresholds within subjects are relatively stable, supported by correlations between the first and final average exit threshold for each subject within each reward environment (short-shallow: $r=0.663$, $p=2.59e-9$; short-steep: $r=0.760$, $p=1.15e-11$; long-shallow: $r=0.599$, $p=1.10e-6$; long-steep: $r=0.312$, $p=1.91e-2$). Finally, to reduce potential confounds of initial learning-related adjustments, we recalculated the average exit threshold for each subject excluding their first 3 exit decisions within each block. The leaving thresholds with and without these first three exit decisions were highly correlated (short-shallow environment $r=0.970$, $p=6.78e-35$; short-steep environment $r=0.977$, $p=7.17e-38$; long-shallow environment $r=0.940$, $p=2.19e-26$; long-steep environment $r=0.969$, $p=1.49e-24$). We recalculated the total change in leaving threshold using the filtered average leaving threshold and the correlations with PCA components 1 and 4 remained significant (component 1 $p=1.04e-2$, component 4 $p=6.44e-3$).

Supplementary Figure 1: Average reward patch exit thresholds within each reward patch

Supplementary Figure 2: Average reward patch exit thresholds grouped by previous reward environment.

Comment 2a:

“Second, the primary behavioral measure which the Authors chose to study (adjustments in the leaving threshold across contexts) is reasonable enough, but it also misses out on other potentially informative relationships. Are dopamine measures (or PCs of dopamine measures) related to the mean threshold?”

Response:

We appreciate this question, which was also raised by Reviewer 2. To address this important gap, we ran additional correlation analyses between the dopamine measures (ROI values and PCA scores) and mean exit threshold and did not find any significant correlations or trends (ROI values minimum p-value of 0.179, PCA scores minimum p-value of 0.305). However, there are positive correlations with the leaving-threshold in the reward environment with the highest average reward rate (short travel time and shallow decay rate) and PCA component 1

score. Individual ROI results revealed positive correlations with $D_{2/3}$ receptor binding potential in the caudate nucleus and ventral striatum (and trend in the putamen) as well as D_1 receptor binding potential in the ventral striatum. We have included the results from these additional analyses in the Supplementary Material.

In the manuscript:

Added to the Supplementary Material (p. 4, second paragraph):

“PCA Component Correlations with Mean Patch-Leaving Thresholds

To aid in interpretation of our threshold change results, we also ran linear regressions with mean patch-leaving threshold (across all four reward environments as well as each individually) as the dependent variable and the four dopamine PET PCA component scores as the independent variables. The mean patch-leaving threshold across all reward environments was not related to the dopamine PCA component scores (complete model $p=0.719$, individual component score p -values >0.3). When looking at the mean leaving thresholds for the individual reward environments, we found that the mean threshold for the reward environment with the higher average reward rate (short travel time and shallow decay rate) was positively correlated with component 1 score (t -stat=2.136, $p=0.0404$) although the complete regression model was not significant ($p=0.141$). There were no significant correlations with the PCA component scores and the mean leaving thresholds for any of the other reward environments (all $p>0.1$).

Individual ROI Correlations with Mean Patch-Leaving Threshold

We ran linear partial correlations between each PET ROI value and our behavioral measures of interest, controlling for age and gender. There are no significant correlations between mean patch-leaving threshold across all environments and the individual PET ROI values (smallest p -value is 0.1794). When looking at the individual reward environment patch-leaving thresholds, there are significant positive correlations between the leaving-threshold for the short-shallow reward environment and $D_{2/3}$ receptor binding potential in the caudate nucleus ($r=0.3141$, $p=0.0455$), ventral striatum ($r=0.3216$, $p=0.0403$), and a trend in the putamen ($r=0.2958$, $p=0.0604$). There is also a trend towards a positive correlation with D_1 receptor binding potential in the ventral striatum ($r=0.2743$, $p=0.0751$). Dopamine presynaptic synthesis capacity is not correlated with any of the individual environment leaving thresholds or with D_1 or $D_{2/3}$ receptor binding potential and the leaving threshold in any of the other reward environments (all $p>0.1$).”

Comment 2b: “What about the proximity to the MVT optimal leaving threshold?”

Response:

The reviewer raises an interesting question about dopamine’s role in the deviation from optimal behavior. To this end, we calculated the absolute value of the difference between average exit threshold in each patch and the optimum leaving threshold according to the MVT for each subject. We then ran partial correlations with the PET measures (individual ROI values

and PCA scores), controlling for age and sex. There were no significant correlations or trends with the PCA scores, but we found trends with deviation from optimal exit threshold and the individual ROI PET values, as noted below.

In the manuscript:

We added the following text to the Supplementary Material (starting at the bottom of p. 7):

“Correlations with MVT-Predicted Optimal Exit Threshold

For each subject, we calculated the absolute value of the difference between their average exit threshold for each patch and the optimal leaving threshold according to the MVT. We then ran linear correlations with the PET PCA and individual ROI values. There were no significant correlations or trends with the PCA scores (minimum p-value of 0.1151). There a negative trend between the deviation from optimal leaving threshold in the short-shallow reward environment and $D_{2/3}$ binding potential in the caudate nucleus ($r=-0.2706$, $p=0.0871$) and ventral striatum ($r=-0.3061$, $p=0.0516$). In addition, there was a positive trend between deviation from MVT optimal leaving threshold in the long-steep environment and dopamine synthesis capacity in the ventral striatum ($r=0.2421$, $p=0.0973$). None of these correlations held up to correction for multiple comparisons.”

Comment 3:

“Third, it seems important to ask whether BMI relates to dopamine or behavior. Indeed, BMI has been found to correlate with reward sensitivity and also D2 receptor availability. It may plausibly also relate to radioligand uptake and signal strength. I would be interested to know whether it explains key variance in the individual difference correlations reported here and also to what degree it relates to foraging behavior in general.”

Response:

The question about BMI is an excellent point given the prior studies noted by the reviewer on dopamine’s role in obesity. To address this, we obtained BMI data from the medical record around the time of the PET scans and included BMI in the multiple regression and partial correlation models with behavioral measures and PET PCA and ROI data. BMI was not associated with foraging behavioral measures including change in exit threshold and change in reaction time. Including BMI in the linear regression model with the PCA components for both of these behavioral measures did not change the results. The individual ROI data trend seen with total change in leaving threshold and $D_{2/3}$ binding receptor was no longer significant when BMI was added as a covariate of no interest, but there was a loss of power due to missing BMI data for 14 of the 43 subjects. These results have been added to the Supplementary Material.

In the manuscript:

Added to the Supplementary Material (p. 8, second paragraph):

“Body Mass Index (BMI) Association with Foraging Behavior and PET Measures

To assess for an impact of BMI on foraging behavior and PET measures, we extracted BMI data from the medical record. Forty-one participants had at least one

BMI measure around the time of their PET scan. Forty-one participants had BMI data at the time of the [¹⁸F]-FDOPA scan, 30 at the time of the [¹¹C]-NNC112 scan, and 29 at the time of the [¹⁸F]-Fallypride scan. BMI values ranged from 19.1 to 33.9 with a median of 25.4. We calculated the mean BMI across all PET scans and included it in a linear regression model with the PCA component scores and behavioral measures of interest. For the total change in leaving threshold, we found that BMI was not associated with behavior ($p=0.659$) and the correlations with components 1 and 4 remained significant at $p<0.05$ with BMI included in the model. Likewise, for the total change in reaction time, BMI was not correlated with behavior ($p=0.599$) and the correlations with components 1 and 4 remained significant at $p<0.05$. Controlling for BMI in the ROI analyses did not change the correlations and trends with D₁ receptor binding potential or presynaptic synthesis capacity, but the correlations with D_{2/3} binding potential no longer met a trend level of significance. However, this is difficult to interpret because the number of subjects dropped from 43 to 29 due to not being able to obtain BMI measurements for all participants.”

Comment 4:

“Fourth, I noticed that the Authors only reported tests of whether PC 1 and 4 relate to individual differences in the change in RT across contexts. Why did the authors only examine components 1 and 4? In the manuscript, they argue that they restricted their analyses to only these components because only 1 and 4 were found to relate to threshold adjustments. Yet, while there may be some conceptual overlap, mean thresholds and mean RTs are also conceptually dissociable. We cannot be certain that behavioral invigoration is one and the same with MVT-based opportunity costs. As such, it seems reasonable to give the other PCs the same treatment in relation to RTs as in relation to patch leaving thresholds.”

Response:

The reviewer raises an excellent question about whether change in reaction time is correlated with the other components not originally tested. To address this, we tested for linear correlations with the change in reaction time and PCA component scores 2 and 3. However, there were no significant correlations or trends (minimum p-value 0.246). We have added this information to the Supplementary Material for completion.

In the manuscript:

Added to the Supplementary Material (bottom of p. 7):

“Given that dopamine may have different effects on reaction time and exit threshold, we ran additional linear correlations between PCA components 2 and 3 and the total change in reaction time. However, there were no significant correlations or trends (minimum p-value of 0.246).”

Comment 5:

“Finally, the main focus of the manuscript, as reflected in the title, is that individual differences in dopamine signaling alters how people trade reward against time costs. As such, the manuscript should also point to a growing literature implicating striatal dopamine signaling in

biasing sensitivity to the benefits versus costs of action. It may be worthwhile to discuss this finding in context of broader theoretical models of how dopamine plays a role both in the learning and performance of actions relative to their respective benefits and costs (e.g., Collins and Frank (2014; Psychological Review), and Möller and Bogacz (2019; PLoS Comp Bio) and also Westbrook and Frank (2018; Curr Opin in Beh Sci) who have considered how such benefits versus costs mechanisms may also drive vigor in rich environments. In so doing, they could make a more explicit connection between the kinds of reward/travel cost tradeoffs studied in foraging tasks and the broader hypotheses by which dopamine signaling is thought to mediate the tradeoff between benefits and costs more generally.”

Response:

The reviewer highlights an important piece of literature that was not included in the initial introduction. We have taken this body of work into account and incorporated it into the manuscript, as highlighted below.

In the manuscript:

Added to the Introduction (p. 5, second paragraph):

“Computational models predict that striatal tonic dopamine encodes the average reward rate of the environment¹⁰ and, along with dopamine receptor activation, plays a role in weighing costs and benefits in the decision to exploit known reward sources or explore for new ones^{11,14,15}.”

(p. 6, first paragraph):

“Increased tonic dopamine is also thought to drive increased rate and vigor of response seen in animal studies¹⁰, as well as modulate how benefits and costs of actions are represented at the time of choice^{11,16,17}.”

(p. 6, second paragraph):

“While there have not been any human studies investigating the role of dopamine D₁ receptors in foraging behavior, computational models and work in animals and human genetics suggest that both D₁ and D₂ receptors are important for decisions that involve weighing costs and benefits and adjusting responses to maximize rewards. Specifically, there is a body of evidence supporting opposing learning effects mediated by D₁ and D₂ receptors facilitating approach and avoidance learning, respectively^{16,17}. Tonic dopamine at the time of choice is thought to modulate the D₁ and D₂-mediated action values to differentially affect representations of benefits and costs.”

Added to the Discussion (p. 21, second paragraph):

“In addition, our results implicating both D₁ and D_{2/3} receptors in foraging behavior adds to prior knowledge about the role of D₂ receptors on foraging behavior³ and suggests that both receptor types are important for adjustments in the foraging patch leaving decision threshold. This supports computational models on the dual actions of dopamine receptors for learning and decision making, such that D₁ receptors are important for learning the benefits of an action, such as staying in a reward patch,

while D_2 receptors play a key role in learning about costs, such as the opportunity cost of lost time^{11,16}. Therefore, both D_1 and D_2 receptor-mediated functions are essential for accurately weighing both benefits and costs to adjust behavior based on changes in the environment.”

References*

*Numbering below corresponds to appearance in 'Response' sections in this letter and not the manuscript.

1. Constantino, S. M. *et al.* *A Neural Mechanism for the Opportunity Cost of Time*. <http://biorxiv.org/lookup/doi/10.1101/173443> (2017) doi:10.1101/173443.
2. Vingerhoets, F. J., Schulzer, M., Ruth, T. J., Holden, J. E. & Snow, B. J. Reproducibility and discriminating ability of fluorine-18-6-fluoro-L-Dopa PET in Parkinson's disease. *J. Nucl. Med.* **37**, 421–426 (1996).
3. Egerton, A., Demjaha, A., McGuire, P., Mehta, M. A. & Howes, O. D. The test-retest reliability of 18F-DOPA PET in assessing striatal and extrastriatal presynaptic dopaminergic function. *NeuroImage* **50**, 524–531 (2010).
4. Dunn, J. T. *et al.* Establishing Test–Retest Reliability of an Adapted [18F]Fallypride Imaging Protocol in Older People. *J. Cereb. Blood Flow Metab.* **33**, 1098–1103 (2013).
5. Kaller, S. *et al.* Test-retest measurements of dopamine D1-type receptors using simultaneous PET/MRI imaging. *Eur. J. Nucl. Med. Mol. Imaging* **44**, 1025–1032 (2017).
6. Chakroun, K., Mathar, D., Wiehler, A., Ganzer, F. & Peters, J. Dopaminergic modulation of the exploration/exploitation trade-off in human decision-making. *eLife* **9**, e51260 (2020).
7. Frank, M. J., Doll, B. B., Oas-Terpstra, J. & Moreno, F. Prefrontal and striatal dopaminergic genes predict individual differences in exploration and exploitation. *Nat. Neurosci.* **12**, 1062–1068 (2009).
8. Behrens, T. E. J., Woolrich, M. W., Walton, M. E. & Rushworth, M. F. S. Learning the value of information in an uncertain world. *Nat. Neurosci.* **10**, 1214–1221 (2007).
9. Jocham, G. *et al.* Reward-Guided Learning with and without Causal Attribution. *Neuron* **90**, 177–190 (2016).
10. Perez, S. M., Elam, H. B. & Lodge, D. J. Increased Presynaptic Dopamine Synthesis Capacity Is Associated With Aberrant Dopamine Neuron Activity in the Methylazoxymethanol Acetate Rodent Model Used to Study Schizophrenia-Related Pathologies. *Schizophr. Bull. Open* **3** (2022).
11. Grace, A. A. Phasic versus tonic dopamine release and the modulation of dopamine system responsivity: A hypothesis for the etiology of schizophrenia. *Neuroscience* **41**, 1–24 (1991).
12. Grace, A. A. Dysregulation of the dopamine system in the pathophysiology of schizophrenia and depression. *Nat. Rev. Neurosci.* **17**, 524–532 (2016).

REVIEWER COMMENTS

Reviewer #1 (Remarks to the Author):

The authors have conducted very careful revisions and have addressed all of the concerns I raised in my previous review, including by providing additional analyses and data. I have also looked at the authors responses to the points raised by the other reviewers, and found the additional analyses provided overall convincing and helpful.

Some typos and additional minor points:

Both at the beginning of the first and last paragraphs of the discussion, the authors prominently emphasize novelty (first study to show, etc.). In my view such statements are not helpful and I would suggest to emphasize the substance of the findings rather than the novelty, as novelty is not a value in itself.

The paper closes by highlighting that the effect found in the study might have wide applicability to disorders ranging from depression to schizophrenia. I know that emphasizing potential clinical applications is common practice in the field, but given that these findings are based on a small sample of healthy volunteers (n=37), I find the highly prominent position of this statement (final sentence of the paper) inappropriate, and would suggest to either remove this statement, substantially weaken it, or move it to another section of the discussion. A related point is that whether this effect relates to time valuation over and above the context of the specific foraging task studied here remains to be shown in future work.

Line 118 and 130 should read “pharmacological”

Line 433 should read “acceleration of response times”

Figure 4c: I suggest to remove the whitening of coefficients between -0.2 and 0.2 , to provide the reader with the full picture of how these components are structured.

Reviewer #2 (Remarks to the Author):

Overall, the authors have done a comprehensive job of responding to comments. The attempt to integrate different PET measures remains a unique strength of the paper. My original primary concern about the stability and interpretability of the PCA results has been reasonably addressed, with the only further comment being that it might be useful after presenting their new data on the stability of the PCA solutions if they added a comment essentially articulating that those analyses support the reliability of the first and second components, but suggest caution in the generalizability of conclusions regarding relations with the 3rd or 4th component.

One thing that surprises me is the new information presented in response to Reviewer 1 about the length of time between assessments is truly substantial, and it is unclear why this long delay was necessary. In some subjects the task was years removed from the scans. The fact that effects remain after covarying for time is encouraging, but it requires careful attention in wording. For instance, the reliability issue is a bit of a challenge because the test-retest reliability data presented for the task is for a very short time span relative to that of the actual study (which in some cases appears to be years). But that is not explicitly acknowledged.

The timing issue warrants greater articulation as limitation than seems acknowledged in the body of the paper. This does not kill the paper's conclusions given that results remain significant after correction for differences in timing, but it also has an impact on how one should interpret the results. Given the delays between task and scanning (on the level of months or even years), results may be interpreted as reflecting a traitwise association, but cannot address the degree to which dopaminergic variables predict behavior at a more precise point in time. The authors treat the issue as something that just adds noise and limits the likely true effect. However, it is not just an issue of noise. What association is present can only really reflect a trait-level association. The degree to which there is additional variance explained by more state-wise changes in dopamine cannot be addressed by in this study design. It would be useful to make that explicit.

I believe that each of the above comments can be addressed with a just a few edits to the current manuscript.

-David Zald

Reviewer #3 (Remarks to the Author):

Ianni and colleagues have satisfied most of my concerns from the prior manuscript. I am glad to know that BMI does not explain key results. I am also glad that they now report trending relationships

between DA PCA measures and mean leaving thresholds as I am sure that many readers will have this question at the top of their mind. I had originally expected them to examine this question when reading the first version myself. So, I think it is important to include relevant analyses in the Supplement, at least.

While most concerns are addressed, there are some remaining considerations that I feel would strengthen the manuscript to more convincingly address any learning effects. Other readers are likely to wonder the same, and while any such effects would not undermine the main conclusions, it seems like a missed opportunity to not assess these a little more.

My biggest concern was regarding the possibility that some individual differences in leaving thresholds might be due to learning effects, and moreover that this covariance might explain some of the correlation between the cross-block change in leaving thresholds and DA PCA scores. In short, the concern was that DA might relate more to individual differences in learning rates rather than the degree to which people shift their exit thresholds across foraging contexts. I am glad the Authors have added some analyses to the Supplement. Specifically, I think that their new test in which they regress the change in leaving thresholds onto DA PCA scores after removing the first three trials affords some confidence that individual differences in the change in patch leaving thresholds are not fully explained by within-block learning effects.

However, it still seems that there might be systematic, within-block changes in the patch leaving threshold as a function of block or previous block. Thus, I am unsure these claims (now in the supplement) are warranted. Eyeballing the moving average patch leaving thresholds suggests that there are systematic within-block changes, and at least these should be acknowledged (or demonstrated more conclusively that there aren't). Such an effect would not undermine their main conclusions but would be informative for theories and models which consider the dual roles of DA on optimizing learning and choice policy, either separately or jointly (e.g. how choice policy should be adapted as a function of learned reward history).

Take, for instance the first figure in the Response letter: the average patch leaving threshold for each participant colored by prior block. In this example, we see, sensibly, that when the prior block was short-shallow (dark blue / navy) most participants tend to reduce their threshold across trials in the other blocks (the pattern is most obvious in the long-steep block). This is consistent with the interpretation that the patch leaving threshold was relatively high in the short-shallow block and there is a kind of learning whereby the threshold progressively decreases when participants enter any other, less rich block type. Also consistent with this interpretation, most participants tend to show a within-block increase in the threshold when the prior block was long-steep (lime green), especially in the short-shallow block.

As such, I would recommend a few additional edits/ analyses. Specifically, I think the Authors could estimate more pointed statistics. For example, they might calculate the difference between patch-leaving threshold by trial and asymptotic patch-leaving threshold in each block (maybe the last 3 trials?), as a function of prior block. According to my eyeball analysis, I expect this statistic to decrease across blocks when the prior block was short-shallow, especially in the long-steep block, and so on. This might show up if they plotted the mean and SEM of the signed or unsigned difference between current and asymptotic patch-leaving threshold, across participants, as a function of prior block (so, in addition to the current Supplemental figures include additional plots showing the statistic as a function of current and prior block).

In any case, I think the Authors should acknowledge that not all participants show stable, asymptotic exit thresholds in all contexts. There are clearly some individuals whose' exit thresholds not only change across trials, but also some who show thresholds that do not asymptote – indeed, they continue to change up until their very last within-block trials.

The Authors could also report the degree to which changes in this statistic across trials correlates with DA measures. In the current revision, the Authors say that the p-values for the regression of between-block leaving thresholds on DA PCs remain < 0.05 , even when you exclude the first three trials of each block. In addition to reporting the estimated effects rather than just the p-values, they could also report a parallel regression across all trials in which they test whether the change in leaving threshold between blocks is regressed on DA PCs, controlling for a more pointed statistic reflecting the within-block change in leaving thresholds. For example, they might control for the within-block change (signed or unsigned) in the leaving threshold from early trials to asymptotic thresholds. If the Authors are right that the between block change in leaving threshold relates to DA PCs, independent of within-block learning effects, then there should still be a significant relationship between between block changes and DA PCs, controlling for within-block changes in this statistic.

Ultimately, based on the Authors' control analyses in the current revision, I am optimistic that their core conclusions will still hold. I am making these recommendations not because I feel strongly that their results reflect learning rather than adaptation to foraging parameters, but because I think a full accounting of DA's effects on learning in addition to their effects on between-block adaptation would provide a more complete picture. Indeed, finding that within-block learning effects explain some or all of the variance in between-block adaptation that might otherwise relate to DA PCs could also be somewhat consistent with their core hypotheses. This outcome would just motivate additional analyses of how DA PCs relate to both within and between block changes in leaving thresholds. Indeed, it would be quite interesting to learn that DA PCs explain both instantaneous between-block changes in leaving thresholds and learning-based adaptation to those thresholds when foraging parameters change. Of course, if these additional analyses do change any conclusions, that should be reflected in the main text.

REVIEWER #1

Comment 1:

“Both at the beginning of the first and last paragraphs of the discussion, the authors prominently emphasize novelty (first study to show, etc.). In my view such statements are not helpful and I would suggest to emphasize the substance of the findings rather than the novelty, as novelty is not a value in itself.”

Response:

The reviewer makes an important suggestion about the specific value this paper adds to the literature. We have taken this into account and have reworded the first paragraph in the discussion as highlighted below.

In the manuscript:

We modified the first sentence of the first paragraph in the discussion (p. 19, last paragraph):

“This study provides valuable insight into the role of dopamine in foraging behavior by measuring multiple facets of the dopamine system in the same individuals who completed a patch foraging task.”

In addition, we adjusted the first sentence of the final paragraph in the discussion (p.26, last paragraph):

“In conclusion, this study revealed a direct correlation between adjustments in foraging behavior and dopamine synthesis capacity and receptor availability in humans.”

Comment 2:

“The paper closes by highlighting that the effect found in the study might have wide applicability to disorders ranging from depression to schizophrenia. I know that emphasizing potential clinical applications is common practice in the field but given that these findings are based on a small sample of healthy volunteers (n=37), I find the highly prominent position of this statement (final sentence of the paper) inappropriate, and would suggest to either remove this statement, substantially weaken it, or move it to another section of the discussion. A related point is that whether this effect relates to time valuation over and above the context of the specific foraging task studied here remains to be shown in future work.”

Response:

We appreciate these thoughtful comments and agree that the manuscript would be stronger with the suggested adjustments, including 1) removing the statement regarding clinical applications and 2) highlighting that whether this effect relates to time valuation over and above the context of the specific foraging task studied here must be resolved by future studies.

In the manuscript:

We have removed the final speculative statement about clinical significance as suggested by the reviewer. The manuscript now concludes with the following sentence (p. 26, last paragraph):

“Our results provide a potential mechanistic explanation for how ACC neural activity underlying foraging decisions (as measured in prior studies) might be modulated by dopamine to enact a change in patch exit threshold based on the specific parameters of the reward environment.”

We have added a sentence to emphasize that further work is needed to see if this effect relates to time valuation over and above the context of the specific foraging task studied here (p. 26, last paragraph):

“Whether dopamine influences time valuation over and above the context of the specific foraging task studied here must be resolved by future studies.”

Comment 3:

“Line 118 and 130 should read “pharmacological”

Line 433 should read “acceleration of response times”

Figure 4c: I suggest to remove the whitening of coefficients between -.2 and .2, to provide the reader with the full picture of how these components are structured.”

Response:

We appreciate these suggestions and have modified the manuscript and Figure 4c accordingly.

In the manuscript:

P. 6, first paragraph:

“A pharmacological study in healthy controls found that administration of a D₂ agonist modulated foraging decisions in poor environments only³.”

P. 6, last paragraph:

“A combined pharmacological and PET study in monkeys revealed that blockade of either D₁ or D₂ receptors reduced the impact of reward and increased delay discounting through a synergistic effect²⁰.”

P. 20, first paragraph:

“Lastly, we found that these two patterns of high dopamine receptor availability throughout the striatum and high mesolimbic dopamine synthesis capacity were also related to acceleration of response times in the reward environment with the highest average reward rate.”

Revised Figure 4c to remove whitening:

REVIEWER #2

Comment 1:

“Overall, the authors have done a comprehensive job of responding to comments. The attempt to integrate different PET measures remains a unique strength of the paper. My original primary concern about the stability and interpretability of the PCA results has been reasonably addressed, with the only further comment being that it might be useful after presenting their new data on the stability of the PCA solutions if they added a comment essentially articulating that those analyses support the reliability of the first and second components, but suggest caution in the generalizability of conclusions regarding relations with the 3rd or 4th component.”

Response:

We are grateful for the positive feedback on our responses to the reviewer’s important comments and glad that the analyses provided have reasonably addressed the reviewer’s primary concern about the stability and interpretability of the PCA results. We agree with the reviewer that it would be useful after presenting the new data on the stability of the PCA solutions to add a comment articulating that the analyses support the reliability of the first and second components and suggest caution in the generalizability of conclusions regarding relations with the 3rd or 4th. We have added to the Results and Discussion accordingly.

In the manuscript:

We added the following information on the stability of the components to the Results (p. 15, last paragraph):

“We assessed reliability of the PCA solution using an independent sample of 26 individuals and found that components 1 and 2 were stable across samples (component 1 between sample $r=0.702$, $p=0.005$; component 2 between sample $r=0.559$, $p=0.038$) whereas components 3 and 4 were not as robustly stable (component 3 between sample $r=0.169$, $p=0.563$; component 4 between sample $r=0.067$, $p=0.820$; see Supplementary Materials for additional details and results from bootstrap analysis).”

In addition, we added a cautionary sentence to the paragraph in the Discussion related to component 4 (p. 23, last paragraph):

“In addition, we suggest caution in the generalizability of conclusions regarding relations with this dopamine component since it did not robustly replicate in our independent sample.”

Comment 2:

“One thing that surprises me is the new information presented in response to Reviewer 1 about the length of time between assessments is truly substantial, and it is unclear why this long delay was necessary. In some subjects the task was years removed from the scans. The fact that effects remain after covarying for time is encouraging, but it requires careful attention in

wording. For instance, the reliability issue is a bit of a challenge because the test-retest reliability data presented for the task is for a very short time span relative to that of the actual study (which in some cases appears to be years). But that is not explicitly acknowledged.”

Response:

The reviewer raises an excellent question about the long delay between PET scan and behavioral data collection for some of the participants. The reason for this delay is that individuals in the present study were recruited from two longer-term dopamine PET protocols that have been ongoing for many years. We aimed to include as many participants as possible in the behavioral study and were unfortunately unable to repeat the PET scans for the individuals with longer length of time between assessments due to IRB constraints. We note the reviewer’s suggestion for careful attention in wording and have adjusted the manuscript accordingly. This is further addressed in the response to the related Comment 3.

In the manuscript:

Added to the Supplementary Materials section on test-retest reliability of foraging behavioral measures (p. 16, last paragraph):

“While the temporal separation between the PET scans and behavioral task for some subjects in our study was quite substantial, it is reassuring that these measures have been shown to be consistent across repeated testing sessions. Future studies are needed to assess longer-term stability of foraging behavioral adjustments.”

Comment 3:

“The timing issue warrants greater articulation as limitation than seems acknowledged in the body of the paper. This does not kill the paper’s conclusions given that results remain significant after correction for differences in timing, but it also has an impact on how one should interpret the results. Given the delays between task and scanning (on the level of months or even years), results may be interpreted as reflecting a traitwise association, but cannot address the degree to which dopaminergic variables predict behavior at a more precise point in time. The authors treat the issue as something that just adds noise and limits the likely true effect. However, it is not just an issue of noise. What association is present can only really reflect a trait-level association. The degree to which there is additional variance explained by more state-wise changes in dopamine cannot be addressed by in this study design. It would be useful to make that explicit.”

Response:

We are grateful that the reviewer raised this very important point. We agree with this distinction that the results presented here reflect a trait-wise rather than state-wise association. We have edited the text to make this explicit for the reader.

In the manuscript:

We adjusted the wording in the limitations section of the Discussion (p. 25, last paragraph):

“Similarly, because PET and behavioral measures were not concurrent, and were in some cases separated by months to years, our findings likely reflect a trait-wise

association and cannot address the degree to which dopaminergic variables predict behavior at a more precise point in time. Future studies using a combined MRI-PET scanner and a displaceable tracer could potentially be used to investigate how state-wise regional dopamine release corresponds to local changes in foraging decision-making.”

REVIEWER #3

Comment 1:

“While most concerns are addressed, there are some remaining considerations that I feel would strengthen the manuscript to more convincingly address any learning effects. Other readers are likely to wonder the same, and while any such effects would not undermine the main conclusions, it seems like a missed opportunity to not assess these a little more.

My biggest concern was regarding the possibility that some individual differences in leaving thresholds might be due to learning effects, and moreover that this covariance might explain some of the correlation between the cross-block change in leaving thresholds and DA PCA scores. In short, the concern was that DA might relate more to individual differences in learning rates rather than the degree to which people shift their exit thresholds across foraging contexts. I am glad the Authors have added some analyses to the Supplement. Specifically, I think that their new test in which they regress the change in leaving thresholds onto DA PCA scores after removing the first three trials affords some confidence that individual differences in the change in patch leaving thresholds are not fully explained by within-block learning effects.

However, it still seems that there might be systematic, within-block changes in the patch leaving threshold as a function of block or previous block. Thus, I am unsure these claims (now in the supplement) are warranted. Eyeballing the moving average patch leaving thresholds suggests that there are systematic within-block changes, and at least these should be acknowledged (or demonstrated more conclusively that there aren't). Such an effect would not undermine their main conclusions but would be informative for theories and models which consider the dual roles of DA on optimizing learning and choice policy, either separately or jointly (e.g. how choice policy should be adapted as a function of learned reward history).

Take, for instance the first figure in the Response letter: the average patch leaving threshold for each participant colored by prior block. In this example, we see, sensibly, that when the prior block was short-shallow (dark blue / navy) most participants tend to reduce their threshold across trials in the other blocks (the pattern is most obvious in the long-steep block). This is consistent with the interpretation that the patch leaving threshold was relatively high in the short-shallow block and there is a kind of learning whereby the threshold progressively decreases when participants enter any other, less rich block type. Also consistent with this interpretation, most participants tend to show a within-block increase in the threshold when the prior block was long-steep (lime green), especially in the short-shallow block.

As such, I would recommend a few additional edits/ analyses. Specifically, I think the Authors could estimate more pointed statistics. For example, they might calculate the difference between patch-leaving threshold by trial and asymptotic patch-leaving threshold in each block (maybe the last 3 trials?), as a function of prior block. According to my eyeball analysis, I expect this statistic to decrease across blocks when the prior block was short-shallow, especially in the long-steep block, and so on. This might show up if they plotted the mean and SEM of the signed

or unsigned difference between current and asymptotic patch-leaving threshold, across participants, as a function of prior block (so, in addition to the current Supplemental figures include additional plots showing the statistic as a function of current and prior block).

In any case, I think the Authors should acknowledge that not all participants show stable, asymptotic exit thresholds in all contexts. There are clearly some individuals whose' exit thresholds not only change across trials, but also some who show thresholds that do not asymptote – indeed, they continue to change up until their very last within-block trials.”

Response:

We are grateful for this thoughtful comment about the dynamics of threshold adjustments reflecting learning during the task. We have adjusted the wording in the Supplementary Materials as the reviewer suggested to highlight the learning effects. Furthermore, we calculated a more pointed statistic as suggested by the reviewer, the signed difference between each leaving threshold and the asymptotic leaving threshold (average threshold over the final three leave decisions). In addition, we ran t-tests on the slope for each individual to test whether the change in average leaving threshold over decisions was significantly different from zero both across the entire reward environment and after excluding the first three leave decisions. We have added these results to the Supplementary Materials and have also included several additional plots highlighting these effects.

In the manuscript:

We re-worded the first paragraph on learning dynamics in the Supplementary Materials (p. 2, last paragraph) to reflect the pattern of learning highlighted by the reviewer.

“Although experimental variables depletion rate and travel time remaining fixed throughout each reward environment, participants were still required to learn about these parameters through sampling when they first entered a new environment. We plotted the running average of the patch leaving threshold over exit decisions within each block and observed that participants did tend to adjust their patch-leaving threshold after the first few decisions before generally settling on a stable threshold for leaving (see Supplementary Fig. 1). However, we also noted individual differences such that some participants and reward environments appeared to have greater learning effects than others, which are also modulated by the previous reward environment that the participant encountered (see Supplementary Fig. 2 and 3).”

We added more pointed statistics of learning as suggested by the reviewer. Specifically, we calculated the difference between patch-leaving threshold by trial and asymptotic patch-leaving threshold (average of exit thresholds for last 3 trials) in each block. Plots of this individual statistic for each participant as well as the group mean and SEM based on the current reward environment as well as the PRIOR reward environment are now included in the Supplementary Materials (p. 5, second paragraph):

“The second approach we used to assess learning effects was to calculate the difference between each individual patch-leaving decision threshold and the asymptotic patch leaving threshold (average of the patch-leaving threshold in the last

three decisions) in each reward environment. We plotted these data for individual participants as well as the mean and standard error across the group based on the current reward environment (see Supplementary Fig. 4 and Fig. 5) and previous reward environment (see Supplementary Fig. 6 and Fig. 7). Participants tend to adjust their exit threshold towards the asymptotic threshold after the first exit decision, appropriately increasing their threshold in the short-shallow reward environment and decreasing their threshold in the environments with the long travel time (Supplementary Fig. 5). The learning effects of prior block appear to last a few more trials in certain cases, specifically in the first block of the experiment and after the long-steep reward environment (Supplementary Fig. 7). Furthermore, the largest initial adjustments in patch leaving threshold appears to occur in the first reward environment of the experiment and followed the short-shallow reward environment.”

We included the following accompanying plots in the Supplementary Materials (p. 6-8):

Supplementary Figure 4: Difference between exit threshold and asymptotic exit threshold (average of last three exit decisions) within each reward patch, plotted by current reward environment. Each color represents an individual participant.

Supplementary Figure 5: Group mean and standard error of the mean (SEM) of the difference between exit threshold and asymptotic exit threshold for each reward environment.

Supplementary Figure 6: Difference between exit threshold and asymptotic exit threshold (average of last three exit decisions) within each reward patch, plotted by PRIOR reward environment. Each color represents an individual participant.

Supplementary Figure 7: Group mean and standard error of the mean (SEM) of the difference between exit threshold and asymptotic exit threshold based on PRIOR reward environment.

Furthermore, we ran additional t-tests on the slope of the average patch leaving threshold over all exit decisions and after excluding the first three leave decisions to formally assess for learning effects across the whole reward block and in the later decisions after excluding the early learning effects. We have added the results to the Supplementary Materials (starting on p. 4, second paragraph):

“To formally test for learning effects, we used two approaches. First, we assessed whether the slope of change of the average patch leaving threshold changed throughout the exit decisions in each reward environment. We used the lm function in R to run a linear regression of average patch leaving threshold over exit decisions within each individual and each reward environment and extracted the coefficient representing the slope of this association. We then ran a t-test across all participants to assess whether the slope was significantly different from zero for each reward environment. We found significant effects of slope in the short-shallow ($t=3.688$, $p=5.178e-4$, $df=55$), long-shallow ($t=-2.7129$, $p=8.886e-3$), and long-steep ($t=-2.6875$, $p=9.506e-3$) reward environments. The slope in the short-steep reward environment was not significantly different from zero ($t=-0.80855$, $p=0.4223$). Assessing longer term learning effects by recalculating the slope of change in average leaving threshold over exit decisions after excluding the first three exit decisions, only the short-shallow reward environment slope remained significantly different from zero ($t=2.361$, $p=2.193e-2$) and the remaining reward environments dropped to trends or non-significant effects (short-steep $t=0.15199$, $p=0.8798$; long-shallow $t=-1.8605$, $p=0.06859$; long-steep $t=-0.66973$, $p=0.5058$). These results suggest that participants

tended to learn early in the block to appropriately adjust their leaving-threshold towards a higher value in the short-shallow reward environment and towards a lower value in the environments with the long travel time. However, the learning effects in the “richest” reward environment (short-shallow) tended to persist beyond the initial three exit decisions.”

Comment 2:

“The Authors could also report the degree to which changes in this statistic across trials correlates with DA measures. In the current revision, the Authors say that the p-values for the regression of between-block leaving thresholds on DA PCs remain < 0.05 , even when you exclude the first three trials of each block. In addition to reporting the estimated effects rather than just the p-values, they could also report a parallel regression across all trials in which they test whether the change in leaving threshold between blocks is regressed on DA PCs, controlling for a more pointed statistic reflecting the within-block change in leaving thresholds. For example, they might control for the within-block change (signed or unsigned) in the leaving threshold from early trials to asymptotic thresholds. If the Authors are right that the between block change in leaving threshold relates to DA PCs, independent of within-block learning effects, then there should still be a significant relationship between between block changes and DA PCs, controlling for within-block changes in this statistic.

Ultimately, based on the Authors’ control analyses in the current revision, I am optimistic that their core conclusions will still hold. I am making these recommendations not because I feel strongly that their results reflect learning rather than adaptation to foraging parameters, but because I think a full accounting of DA’s effects on learning in addition to their effects on between-block adaptation would provide a more complete picture. Indeed, finding that within-block learning effects explain some or all of the variance in between-block adaptation that might otherwise relate to DA PCs could also be somewhat consistent with their core hypotheses. This outcome would just motivate additional analyses of how DA PCs relate to both within and between block changes in leaving thresholds. Indeed, it would be quite interesting to learn that DA PCs explain both instantaneous between-block changes in leaving thresholds and learning-based adaptation to those thresholds when foraging parameters change. Of course, if these additional analyses do change any conclusions, that should be reflected in the main text.”

Response:

The reviewer raises an important question about whether the association between the dopamine components and the between-block threshold adjustments can be explained by within-block learning dynamics. As suggested, we have added effect sizes (t-statistics and degrees of freedom) for the regression analysis testing effects of PET component scores on total change in leaving threshold after excluding the first three leave decisions. In addition, we took two approaches to test whether within-block learning dynamics explain the associations between dopamine components and between-block adjustment in leaving threshold. First, we tested whether including the slope of the average leaving threshold within each block as a control covariate affected the association between PET PCA components 1 and 4 and the between-block change in leaving threshold. Second, as suggested by the reviewer, we

calculated the difference between the first leaving threshold and the asymptotic average leaving threshold and used that value for each reward environment as a control covariate in partial regression analyses with PCA components 1 and 4 and the between-block change in leaving threshold. We found that the correlation between PCA component 1 and the between-block change in leaving threshold remained significant even after controlling for the within-block learning parameters. The correlation with PCA component 4 and between-block change in leaving threshold remained significant when controlling for within-block learning in the short-shallow and long-shallow reward environments but dropped to a trend when controlling for learning in the long-steep reward environment and was no longer significant when controlling for learning in the short-steep reward environment. We have added these results to the Supplementary Materials but did not find that they changed our overall conclusions and thus we did not add them to the main text.

In the manuscript:

Added to the Supplementary Materials p. 12-14:

“PCA Component Correlations with Leaving Threshold Dynamics

To assess whether PET PCA component associations with change in leaving threshold reflects choice policy rather than initial learning dynamics, we recalculated the total change in leaving threshold after excluding the first three leave decisions. We found that the correlations with change in leaving threshold and PCA components 1 and 4 remained significant using the filtered threshold values (component 1 t-stat=2.72, $p=1.04e-2$; component 4 t-stat = 2.92, $p=6.44e-3$, degrees of freedom=32). In addition, we ran partial correlations controlling for within block learning dynamics to assess whether the associations between PET PCA components 1 and 4 and the total change in patch leaving threshold between the “rich” and “poor” environment were independent from within-block learning effects.

First, we used the slope of the average leaving threshold throughout the entire reward environment block as a control variable. We ran separate partial correlations controlling for the slope in each of the reward environments. We found that the correlation between PCA component 1 and the total change in patch leaving threshold remained significant even after controlling for the slopes in each of the reward environments (short-shallow: $r=0.4114$, $p=0.0127$; short-steep: $r=0.4811$, $p=0.0030$; long-shallow: $r=0.3954$, $p=0.0170$; long-steep: $r=0.3659$, $p=0.0282$). The correlation between PCA component 4 and the total change in patch leaving threshold remained significant when controlling for the slope in the short-shallow ($r=0.3568$, $p=0.0327$) and long-shallow ($r=0.3969$, $p=0.0165$) reward environments, but dropped to a trend when controlling for the slope in the long-steep reward environment ($r=0.2898$, $p=0.0864$) and was no longer significant after controlling for the slope in the short-steep reward environment ($r=0.1951$, $p=0.2542$).

Second, we took a complimentary approach to control for within-block learning dynamics by calculating the difference between the first exit threshold and the asymptotic average exit threshold (average of last three exit thresholds) in each block. We then included this signed threshold difference as a control parameter in partial regression analyses measuring the correlation between PET PCA components 1

and 4 and the total change in patch leaving threshold between the “rich” and “poor” reward environments. Consistent with the slope approach reported above, we found that the correlation between PCA component 1 and the total change in patch leaving threshold between the “rich” and “poor” environments remained significant after controlling for the within-block change in threshold in the short-shallow ($r=0.4496$, $p=0.0059$), short-steep ($r=0.4277$, $p=0.0093$), and long-shallow ($r=0.4007$, $p=0.0154$), and long-steep reward environments ($r=0.3453$, $p=0.0392$). For PCA component 4, the total change in patch leaving threshold remained significant after controlling for the within-block change in threshold in the short-shallow ($r=0.3710$, $p=0.0259$) and long-shallow ($r=0.3738$, $p=0.0247$) reward environments, but dropped to a trend when controlling for within-block threshold change in the long-steep reward environment ($r=0.3193$, $p=0.0577$) and was no longer significant when controlling for within-block change in threshold in the short-steep reward environment ($r=0.2231$, $p=0.1910$). Overall, these results suggest that the dopamine PET principal components primarily explain between-block changes in leaving threshold. However, the pattern of dopamine variation in component 4 may also be important for within-block learning, specifically in the reward environments with steep depletion rates.”

REVIEWERS' COMMENTS

Reviewer #3 (Remarks to the Author):

The authors have addressed all my comments and I believe the paper is stronger.